

# A bias-corrected CMIP5 dataset for Africa using CDF-t method. A contribution to agricultural impact studies.

Adjoua Moise Famien[1,2], Serge Janicot[2], Abe Delfin Ochou[1], Mathieu Vrac[3], Dimitri Defrance[2],
Benjamin Sultan[2], and Thomas Noël[4]

[1]Université Félix Houphouët Boigny, LAPAMF-UFR SSMT, 22 BP 582, Abidjan 22, Ivory Coast
[2]UMR 7159 LOCEAN, Sorbonne Universités UPMC-CNRS-IRD-MNHN, Paris, France
[3]LSCE-IPSL, CNRS/CEA/UVSQ, Centre d'Études de Saclay, Orme des Merisiers, Gif-sur-Yvette, France
[4]Climate Data Factory, Paris, France

*Correspondence to:* Serge Janicot (serge.janicot@locean-ipsl.upmc.fr)

**Abstract.** The objective of this paper is to present a new data set of bias-corrected CMIP5 global climate models (GCMs) daily data over Africa. This dataset was obtained in using the Cumulative Distribution Function Transform (CDF-t) method, a method that has been applied on several regions and contexts but never on Africa. Here CDF-t is used over the period 1950-2099 combining historical runs and climate change scenarios on 6 variables, precipitation, mean near-surface air temperature, near-surface maximum air temperature, near-surface minimum air temperature, surface down-welling shortwave radiation, and wind speed, which are critical variables for agricultural purposes. Evaluation of the results is carried out over West Africaon a list of priority users-based metrics that was discussed and selected with stakeholders and on simulated yield using a crop model simulating maize growth. Bias-corrected GCMs data are compared withanother available dataset of bias-corrected GCMs, and the impact of three different reference datasets on bias-correctionsis also examined in details. CDF-t is very effective in removing the biases and in reducing the high inter-GCMs scattering. Differences with other bias-corrected GCMs data are mainly due to the differences between the reference datasets. This is particular true for surface down-welling shortwave radiation, which has impacts in terms of simulated maize yields. Projections of future yields over West Africa have quite different levels, depending on bias-correction method used, but they all show a similar relative decreasing trend over the 21st century.

## 1 Introduction

Global (GCMs) and regional (RCMs) climate models are used to produce projections of future climates driven by various types of greenhouse gas emission scenarios. The last Coupled Model Intercomparison Project (CMIP ; Meehl et al., 2000), CMIP5, provides simulations for preindustrial period ($CO_2$ concentration at a level of 280 ppm), historical period (1860-2005; including real evolutions of $CO_2$ and other greenhouse gas concentrations, anthropogenic and volcanoes eruptions aerosols



contents, solar activity), and future climate projections based on different $CO_2$ emission trajectory scenarios, Representative Concentration Pathways RCPx.x (Moss et al. 2010; x.x corresponding to the radiative forcing in $W\,m^{-2}$ in 2100), RCP2.6, RCP4.5, RCP6.0 and RCP8.5 (Taylor et al., 2012).

Scientific communities working on evaluation and modelling of climate change impacts (in terms of crop yields, water

resources, health, etc) are increasingly using these simulations outputs either to compute related impact metrics or to run impact models. However robust biases are still present in climate models due to ill-defined processes and associated parametrisations, leading to biased statistical distributions of simulated physical and dynamical variables (e.g., Vrac and Friederichs, 2015). Then statistical bias-corrections are necessary before variables being introduced in impact models simulations (Vrac et al., 2016). GCMs and RCMs output data are adjusted to statistical distributions similar to observations ones used as reference data.

The use of different bias-correction methods based on different reference data sets contributes to the total uncertainty in climate projections and can contribute in some contexts more than the use of different GCMs or RCMs (Iizumi et al., 2017). So using multiple bias-correction technics and reference data sets can be recommended. For instance, a bias-correction of a subset of 5 GCMs of the CMIP5 database was realised at a global scale through the ISIMIP project (Hempel et al., 2013), the first Inter-Sectorial Impact Model Intercomparison Project[1]. These corrections were applied at a daily scale from 1 January 1950

to 31 December 2099, on Historical and all RCPs scenarios on 5 GCMs at a $0.5°$ x $0.5°$ grid, using WFD data as observation-based reference. More recently a ISIMIP2b bias-correction using an improved reference data set EWEMBI has been realized on 4 out of the 5 same CMIP5 GCMs data, and the results have been compared to the bias-corrected ISIMIP/WFD data (Lange, 2017a). Significant differences have been highlighted that are closely related to differences between WDF and EWEMBI data.

Modelling climate suffers high and robust biases over Africa. For instance, GCMs display major sea surface temperature

warm biases in the equatorial Atlantic inducing a systematic southward shift of the summer monsoon over West Africa, and little evolution has been shown between CMIP3 and CMIP5 simulations (Roehrig et al., 2013). Biases have also been identified elsewhere in Africa, for instance over Central Africa (Washington et al., 2013) or East and South Africa, even if inconsistencies also exist between the different available data sets of reanalysis (Brands et al., 2013). So introducing bias-correction processes is mandatory over this continent before addressing any climate change impacts issue.

The objectives of this paper are to present and evaluate bias-corrected GCMs data obtained by performing the Cumulative Distribution Function Transform (CDF-t) method over Africa, and quantify the sensitivity to different reference datasets of the bias-corrected data, in particular in terms of simulated crop yields. It is a contribution to the AMMA-2050[2] project, centred on West Africa, that has the following goals : (i) significantly improved scientific understanding of climate variability and change across Africa and the impact of climate change on specific development decisions; (ii) demonstration of flexible methods for

integrating improved climate information and tools in specific decision-making contexts; (iii) improved medium-to long-term (5-40 year) decision-making, policies, planning and investment by African stakeholders and donors.

Then, bias-correction has been applied on daily data of six variables critical for these types of impact: precipitation (pr), mean near-surface air temperature (tas), near-surface maximum air temperature (tasmax), near-surface minimum air tempera-

---

[1]https://www.isimip.org/

[2]http://www.amma2050.org/





ture (tasmin), surface down-welling shortwave radiation (rsds), and wind speed (wind). The bias-correction has been performed using the Cumulative Distribution Function-transform (CDF-t ; Michelangeli et al., 2009), a method that has been widely used and validated for various variables and in various contexts (e.g., Kallache et al., 2011; Vrac et al., 2012; Lavaysse et al., 2012; Vautard et al., 2013; Vrac and Friederichs, 2015; Vrac et al., 2016), including tropical areas (Oettli et al., 2011; Vigaud et al., 2013). These corrections have been applied on 29 GCMs over 1950-2005 period and different RCP2.6, RCP4.5 and RCP8.5

2006-2099 projections. The observation-based reference dataset used for bias-corrections is WFDEI, the WATCH Forcing Data (WFD; Weedon et al., 2011) methodology applied to ERA-Interim data, for the period from 1 January 1979 to 31 December 2013 on a 0.5° x 0.5° grid (Weedon et al., 2014).

Section 2 presents these reference data used to evaluate the 29 CMIP5 GCMs. A first intercomparison of WFD, WFDEI and EWEMBI is presented is terms of mean seasonal fields over West Africa. In Section 3 the CDF-t bias-correction method

is shortly presented. Then tests are carried out over 1979-2013 to evaluate the sensitivity of the corrections to the calibration period. In section 4, the evaluation of the CDF-t bias-correction is detailed over West Africa, first on mean seasonal fields, then on daily-based metrics. CDF-t bias-corrected GCM data are also compared with ISIMIP/WFD bias-corrected data for the 5 GCMs used in ISIMIP. The significant impact induced by some improvements introduced in WFDEI data will be shown. CDF-t outputs will be also compared to products from EWEMBI. To go further into this evaluation, a crop model has been

used to test the impact on simulated crop yields (specifically a local maize cultivar) of bias-corrections data with one GCM and of the three reference data sets, used as forcing data, in order to see how such crop simulation can integrate (non-linearly) these corrections. A sensitivity analysis to individual forcing variables (temperature, precipitation and surface down-welling shortwave radiation) is also presented. Finally some analyses of the bias-correction impact on crop simulations in the context of RCP8.5 climate change projections are shown. Conclusions are given in section 5.

## 20  2   Climate input data

The AMMA-2050 data set comprises bias-corrected daily data for the variables precipitation, mean near-surface air temperature, maximum air temperature and minimum air temperature, surface down-welling shortwave radiation, and wind speed. It covers the domain 20° W-55° E/40° S-40° N, including the whole Africa. In this paper, results are presented for West Africa (20° W-20° E/0-25° N) for northern summer. Similar results but for northern spring are provided in Supplementary

Information.

### 2.1   Simulations

We use daily data extracted from the CMIP5 archive, covering the period from 1 January 1950 to 31 December 2099. Based on availability of daily data, it comprises 29 GCMs for the 1950-2005 historical period and RCP8.5 2006-2099 projection, 27 GCMs for RCP4.5 projection and 20 GCMs for RCP2.6 projection (See Table 1 for more details). Only one run has

been used for each GCM. These "raw" data have been interpolated on the 0.5° x 0.5° grid of WFDEI for an easier comparison

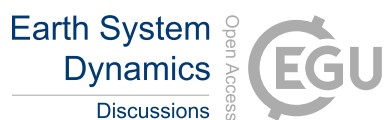


**Table 1.** List of available CMIP5 models used for historical and RCP simulations. The 5 GCMs also used in ISIMIP are in italics. A number in each column is the number of ensemble member used in this work. Zero indicates that no run was used. The last line shows the total number of run used for each simulation.

| Modelling Centre (or Group) | CMIP5 Models | Resolution (lat x lon x lev) | Historical | Simulations RCP2.6 | RCP4.5 | RCP8.5 |
|---|---|---|---|---|---|---|
| Commonwealth Scientific and Industrial Research Organization (CSIRO) and Bureau of Meteorology (BOM), Australia | ACCESS1-0 | 1.25° x 1.875° x 38 | 1 | 0 | 1 | 1 |
| | ACCESS1-3 | | 1 | 0 | 1 | 1 |
| Beijing Climate Center, China Meteorological Administration | bcc-csm1-1 | 1.875° x 1.875° x 16 | 1 | 1 | 1 | 1 |
| | bcc-csm1-1-m | | 1 | 1 | 1 | 1 |
| College of Global Change and Earth System Science, Beijing Normal University | BNU-ESM | 2.81° x 2.81° x 26 | 1 | 1 | 1 | 1 |
| Canadian Centre for Climate Modelling and Analysis | CanESM2 | 2.790° x 2.81° x 35 | 1 | 1 | 1 | 1 |
| Centro Euro-Mediterraneo per I Cambiamenti Climatici | CMCC-CESM | 3.443° x 3.75° x 39 | 1 | 0 | 0 | 1 |
| | CMCC-CM | 0.748° x 0.75° x 31 | 1 | 0 | 1 | 1 |
| | CMCC-CMS | 3.711° x 3.75° x 95 | 1 | 0 | 1 | 1 |
| Centre National de Recherches Météorologiques/Centre Européen de Recherche et Formation Avancée en Calcul Scientifique | CNRM-CM5 | 1.4° x 1.4° x 31 | 1 | 1 | 1 | 1 |
| Commonwealth Scientific and Industrial Research Organization in collaboration with Queens land Climate Change Centre of Excellence | CSIRO-Mk3-6-0 | 1.875° x 1.875° x 18 | 1 | 1 | 1 | 1 |
| NOAA Geophysical Fluid Dynamics Laboratory | GFDL-CM3 | 2° x 2.5° x 48 | 1 | 1 | 1 | 1 |
| | GFDL-ESM2G | 2° x 2.5° x 24 | 1 | 1 | 1 | 1 |
| | GFDL-ESM2M | 2° x 2.5° x 24 | 1 | 1 | 1 | 1 |
| Met Office Hadley Centre (additional HadGEM2-ES realizations contributed by Instituto Nacional de Pesquisas Espaciais) | HadGEM2-AO | 1.25° x 1.875° x 38 | 1 | 1 | 1 | 1 |
| | HadGEM2-CC | 1.25° x 1.875° x 38 | 1 | 0 | 1 | 1 |
| | HadGEM2-ES | 1.25° x 1.875° x 38 | 1 | 1 | 1 | 1 |
| Institute for Numerical Mathematics | Inmcm4 | 1.5° x 2° x 21 | 1 | 0 | 1 | 1 |
| Institut Pierre-Simon Laplace | IPSL-CM5A-LR | 1.9° x 3.75° x 39 | 1 | 1 | 1 | 1 |
| | IPSL-CM5A-MR | 1.25° x 2.5° x 39 | 1 | 1 | 1 | 1 |
| | IPSL-CM5B-LR | 1.9° x 3.75° x 39 | 1 | 0 | 1 | 1 |
| Atmosphere and Ocean Research Institute (The University of Tokyo), National Institute for Environmental Studies, and Japan Agency for Marine-Earth Science and Technology | MIROC5 | 1.4° x 1.4° x 40 | 1 | 1 | 1 | 1 |
| Japan Agency for Marine-Earth Science and Technology, Atmosphere and Ocean Research Institute (The University of Tokyo), and National Institute for Environmental Studies | MIROC-ESM | 2.8125° x 2.8125° x 80 | 1 | 1 | 1 | 1 |
| | MIROC-ESM-CHEM | | 1 | 1 | 1 | 1 |
| Max-Planck-Institut für Meteorologie (Max Planck Institute for Meteorology) | MPI-ESM-LR | 1.8653° x 1.875° x 47 | 1 | 1 | 1 | 1 |
| | MPI-ESM-MR | 1.8653° x 1.875° x 95 | 1 | 1 | 1 | 1 |
| | MRI-CGCM3 | 1.12148° x 1.125° x 48 | 1 | 1 | 1 | 1 |
| Meteorological Research Institute | MRI-ESM1 | | 1 | 0 | 0 | 1 |
| Norwegian Climate Centre | NorESM1-M | 1.9° x 2.5° x 26 | 1 | 1 | 1 | 1 |
| **Total** | 29 | | 29 | 20 | 27 | 29 |



with observation, by a bilinear approach for temperatures and wind, and by a "nearest neighbour" approach for precipitation. Then, bias-corrected data are available on the 0.5° x 0.5° grid.

## 2.2 Reference observation data sets

The observation-based reference dataset is critical for the correction of GCM biases, in particular when corrections are applied on daily data. The reference dataset must also have a global coverage with a regular grid, what may induce large uncertainties in void in-situ data areas as in Africa. So we used the opportunity of the availability of WFD, WFDEI and EWEMBI reference datasets to compare each other, and to compare bias-corrected (with WFD) ISIMIP data with bias-corrected (with WFDEI) AMMA-2050 data.

The WFD dataset (Weedon et al., 2011) is a combination of ERA-40 daily reanalysis of the European Centre for Medium-Range Weather Forecasts (ECMWF) at a grid resolution of 2.5° and the Climate Research Unit (CRU) TS2.1 dataset that provides observed time series of monthly variations in the climate on a resolution grid of 0.5° . A correction for monthly mean rainfall is included using the Global Precipitation Climatology Centre (GPCC) version 4 dataset (Hagemann et al., 2011). The WFD data are available over the period 1958-2001 on a 0.5° grid over land area points. We will use the WFD dataset over 1979-2001.

WFDEI, an improved version of WFD, has been produced based on ERA-Interim reanalysis, over the period from 1 January 1979 to 31 December 2013 on a 0.5° x 0.5° grid (Weedon et al., 2014). Improvements come from the 4D-var data assimilation system with 6h windows in ERA-Interim instead of 3D-var in ERA-40. Compared to ERA-40, ERA-Interim uses a more extensive suite of satellite, atmospheric soundings and surface observations and provides substantial improvement in surface meteorological variables (Dee et al., 2011), in particular with a new aerosol loading distributions and corrections for downward shortwave fluxes (leading in particular to larger average WFDEI values over Sahara and northern Africa) leading to less bias compared to globally distributed observations. ERA-Interim has also a reduced Gaussian grid spectral model resolution of T255 instead of T159 for ERA-40, leading to data much "closer" to the regular 0.5° x 0.5° spatial resolution and to the elevation distribution used for WFDEI. We will use the WFDEI dataset over 1979-2013.

More recently, the EWEMBI data set has been produced within ISIMIP (Lange, 2016, 2017b). Data sources of EWEMBI are ERA-Interim data, WFDEI, eartH2Observe forcing data (E2OBS; Dutra, 2015) and NASA/GEWEX Surface Radiation Budget data (SRB; Stackhouse Jr et al., 2011). Significant differences have been highlighted between WFD-based and EWEMBI-based bias-corrected data that are closely related to similar improvements from WDF to EWEMBI data. We will use the EWEMBI dataset over 1979-2013.

## 2.3 Intercomparison of WFD, WFDEI and EWEMBI on mean seasonal fields over West Africa

In the following, to reduce the number of figures, most of the results are presented only for the summer season, July-September (JAS), which is the main rainy season over the Sahel. Similar computations have been performed over the other seasons, especially over spring, which is the main rainy season over the Guinean Coast, and some of the results will be commented.





**Figure 1.** Summer climatology from different observations datasets (WFD, WFDEI and EWEMBI) : (a-c) for near-surface temperature (°C) over 1979-2001, (d-f) for precipitation rate ($mm\,day^{-1}$) over 1979-2001 period, (g-i) for solar radiation ($W\,m^{-2}$) over 1984-2001 period and difference between WFD (j) over 1984-2007, WFDEI (k) over 1984-2007, EWEMBI (l) over 1984-2001 and SRB solar radiation.

Fig.1 presents the July-September mean seasonal fields of WFD, WFDEI and EWEMBI for tas, pr and rsds. Regarding tas, the mean fields of the three reference data sets are very close, showing the set-up in northern spring and summer of the high temperature area associated to the Saharan and Saudi Arabia heat lows. Regarding pr, the seasonal fields are also very close, showing the seasonal migration of the Inter-Tropical Convergence Zone (ITCZ) between spring and summer. Local maxima associated to highlands like Fouta Jalon or Cameroon mountains are also clearly highlighted. Regarding rsds, more differences are evident between the 3 reference data sets. The mean seasonal fields show similar patterns with low values within the ITCZ





area due to the high cloud coverage and high values over the Sahara due to low moisture and cloud coverage, but the range of values are quite different. Over ITCZ WFD rsds values are the weakest and EWEMBI values the highest. Over the Sahara WFD values are also the weakest but WFDEI values are a bit higher than for EWEMBI. In the remaining panels, anomalies are produced in respect to SRB data. Compared to SRB, EWEMBI data are very similar that is logical since SRB data was used to correct ERA-Interim. WFDEI has moderate negative biases in the ITCZ area and weak positive biases over the Sahara, while WFD has high negative biases over the whole area.

## 5   3   The CDF-transform bias-correction

### 3.1   The CDF-t method

In this work, we use the CDF-t method developed by Michelangeli et al. (2009) to adjust climate models. It consists in matching the CDF of a climate variable simulated by a model (here the GCM) to be corrected at the CDF of this observed variable (here WFDEI) through a mathematical function. CDF-t is a variant of the "quantile mapping" method (Déqué, 2007).

But contrary to "quantile mapping", CDF-t first constructs the CDF of the observations in the future through the mathematical transformation before doing the mapping with the CDF of the models. So, if we assume that $F_{obs,fut}$ is the CDF of the observations in the future, $F_{obs,fut}$ is given by the relation:

$$F_{obs,fut} = F_{obs,cal} \left[ F_{mod,cal}^{-1} [F_{mod,fut}] \right] \tag{1}$$

Where $F_{obs,cal}$ and $F_{mod,cal}^{-1}$ are respectively the CDF of the observation and the inverse CDF of the model over the calibration

period; $F_{mod,fut}$ represents the CDF of the model to be corrected.

More details on the CDF-t method can be found in (Vrac et al., 2012, 2016).

### 3.2   Application

This CDF-t approach has been applied to 5 out of the 6 variables (tas, tasmax, tasmin, rsds and wind) over the period 1950-2099 (historical and RCP2.6, RCP4.5, RCP8.5 runs). For precipitation (pr), an updated CDF-t approach has been used,

referred to as "Singularity Stochastic Removal" (SSR), addressing also rainfall occurrence and intensity issues (see Vrac et al., 2016, for more details). CDF-t has been applied month by month to take in account the strong seasonality over Africa. It has also been applied using a moving window to smooth discontinuities (Vrac et al., 2016). CDF-t preserves any long-term trend in the GCMs data. GCMs data have been interpolated to WFDEI grid before being bias-corrected.

Examples of CDF-t bias-correction applied on mean West Africa daily precipitation data for the five GCMs used in

ISIMIP are shown (Fig.S1). It is represented in terms of cumulative distribution function. The distributions of raw GCM data are clearly different from the WFDEI data. Some of them show more low precipitation values in GCMs than in WDFEI while others have more low precipitation values.The CDF-t bias-correction appears very effective as WFDEI and bias-corrected GCM data distribution are closely superimposed.





### 3.3 Sensitivity of the correction to the calibration period over West Africa

Before applying the CDF-t correction through the moving window process over 1950-2099, every GCM has to be calibrated over a reference period. In order to have a calibration data set as representative as possible of the variability of the various variables, especially precipitation, the time period 1979-2013 has finally been used for calibration of the bias-correction method. However the sensitivity to the calibration period has been explored over West Africa by testing it on two sub-periods, 1979-1996 and 1996-2013, to prevent any over-estimation of the bias-correction performance. This has been performed on the five GCMs used in ISIMIP, and it is morespecifically shownon IPSL-CM5A-LR model in summer for tas, pr and rsds (Supplementary Information).

Three calibration periods have been tested: 1979-1996, 1996-2013 and 1979-2013 (see Fig.S2). First, it is clear that the bias-correction is powerful to remove the cold bias of the raw data. Second, the positive trend present in the raw data over the period 1979-2013, as in WFDEI but with a weaker range, is preserved after the bias-correction. This is probably due to the dry bias of precipitation over the Sahel in raw data that induces a higher sensitivity to the impact of anthropogenic global warming over the period than in observations. Third, the effect of the calibration period is clear. By using the calibration period 1979-1996, the remaining bias of corrected data is near zero and is weakly positive over 1997-2013, while by using the calibration period 1996-2013, the remaining bias of corrected data is near zero and is weakly negative over 1979-1995. Using the calibration period 1979-2013, the remaining bias is overall very weak and in average near zero. Similar tests have been carried out for the variables pr and rsds, and for the other seasons, with similar conclusions. So, while it can be thought that using the whole observational period to calibrate the bias-correction process may lead to over-estimation of the fit between observations and bias-corrected data, it provides in fact a more robust correction. So we keep the whole period 1979-2013 for calibration.

## 4 Results

### 4.1 User-based metrics and diagnostics

A list of priority metrics has been established between scientists and stakeholders involved in AMMA-2050. We are presenting results based on some of these metrics related to the three variables, precipitation (pr), near-surface air temperature (tas) and surface down-welling shortwave radiation (rsds). These metrics are:

- the seasonal mean for pr, tas and rsds,

- the mean time-latitude annual cycle over (15° W-15° E) for pr, tas and rsds,

- the 95[th] percentile of daily values for tas,

- the number of days with tas > 30° C,

- the 95[th] percentile of daily values for pr,



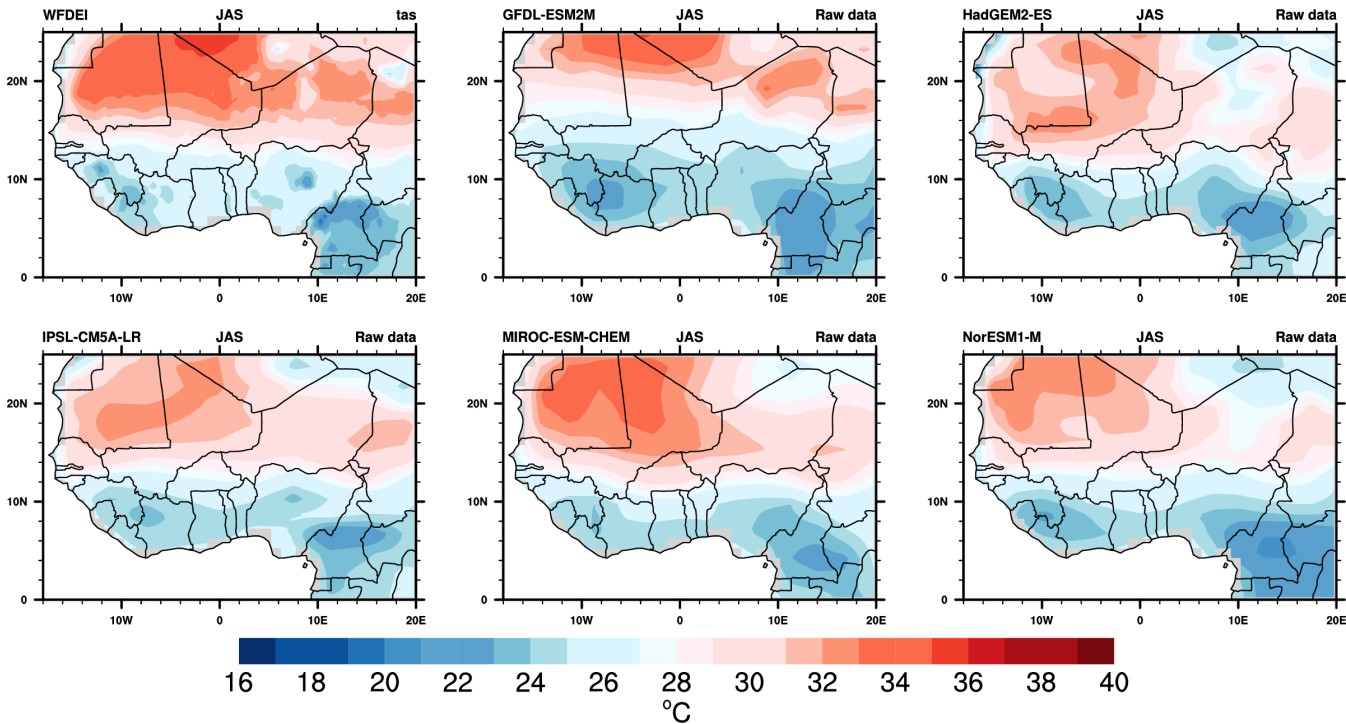

**Figure 2.** Mean surface temperature (° C) from WFDEI and 5 CMIP5 models raw data field over period 1979-2001

- the number of wet days ($\mathrm{pr} > 1 \, \mathrm{mm} \, \mathrm{day}^{-1}$),

- the number of days with $\mathrm{pr} > 10 \, \mathrm{mm} \, \mathrm{day}^{-1}$,

- the number of dry days ($\mathrm{pr} < 1 \, \mathrm{mm} \, \mathrm{day}^{-1}$),

- the 95[th] percentile of the duration of consecutive dry days sequences.

5 **4.2 Mean seasonal fields over West Africa**

Regarding tas, Fig.2 presents the mean JAS temperature fields over Africa for WFDEI data, biases from WFDEI of raw data from the five GCMs used in ISIMIP, and biases of CDF-t bias-corrected GCM data. Fig.3 shows the Taylor diagrams (see Taylor, 2001) computed on JAS over the Sahel and Guinea areas, for (i) the 29 raw and bias-corrected GCM data compared to WFDEI data, (ii) similar but for the five GCMs used in ISIMIP in terms of raw data, CDF-t bias-corrected data and ISIMIP bias-corrected data, (iii) similar except it is evaluated against EWEMBI data. WFD data are also plotted in these Taylor diagrams. Note that the Taylor diagrams provide three statistics: the correlation coefficient between the tested field and the reference field (related to the azimuthal angle), the normalized standard deviation of the tested field in respect to the standard deviation of the reference field (proportional to the radial distance from the origin), and the centred root mean square difference between the tested field and the reference field (proportional to the distance from the REF point on the x-axis; grey curves from 1 to





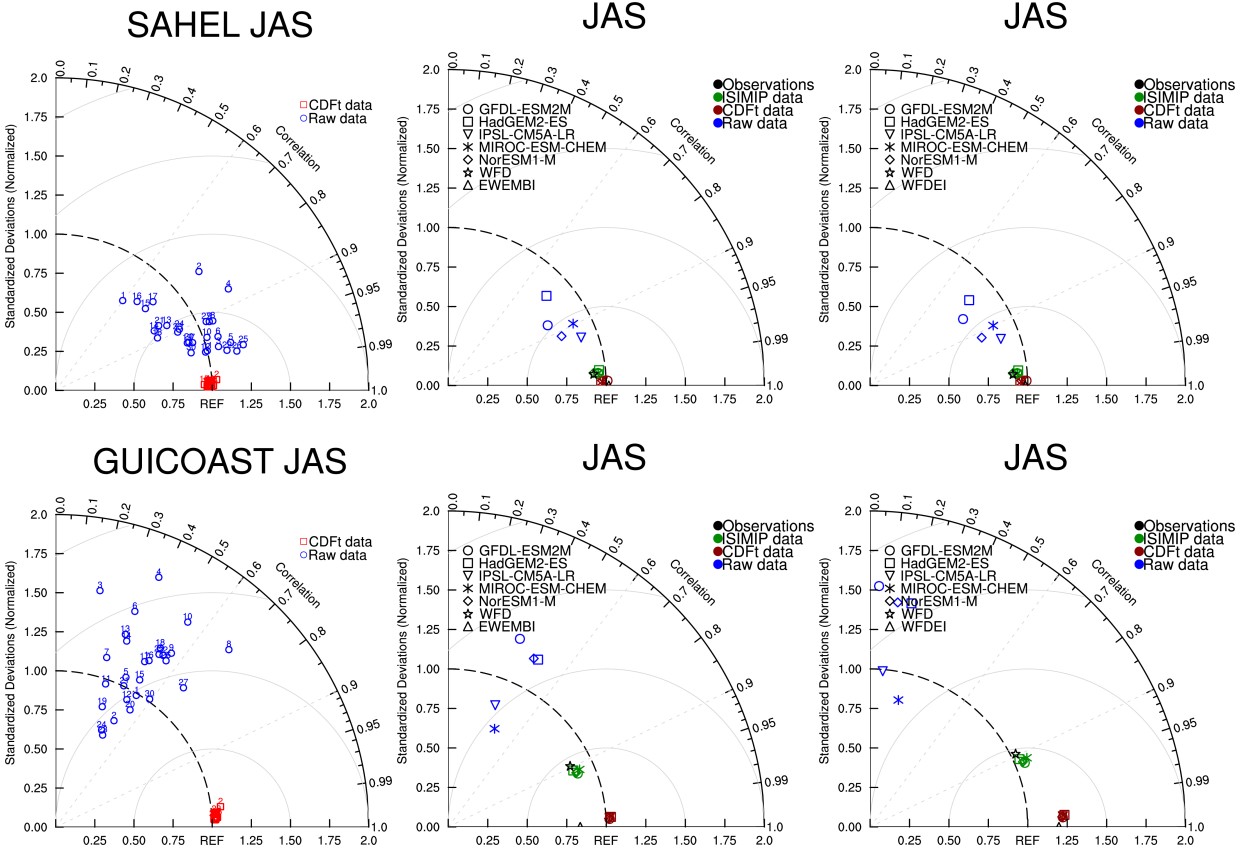

**Figure 3.** Taylor diagrams relative to the mean of surface temperature from 29 individual models and multimodel (1st column) and 5 five models (2nd and 3rd columns) from results involved in CMIP5 historical experiment over 1979-2001 period. Two areas are considered for temperature fields : Sahelian Box (SAHEL : 18° W-10° E ; 10° N-20° N), Guinean box (GUICOAST : 18° W-10° E ; 3° N-10° N). Data are compared to WFDEI data (1st and 2nd columns) and to EWEMBI (3rd column). Taylor diagrams provide three statistics: the correlation coefficient between any tested field and the reference field (angle), the normalized standard deviation of the tested field in respect to the reference observation (x-axis and y-axis), and the centred root mean square difference between the tested field and the reference field (grey circles from 1 (with the lowest radius) to 4 (the highest radius)).

4). As the mean of the fields are subtracted out before computing their second-order statistics, these diagrams do not provide information about overall biases but characterizes biases associated to centred pattern errors. Hence maps of Fig.2 and Taylor diagrams of Fig.3 provide complementary bias information.

Fig.2 shows that globally, the GCMs capture well the spatial structure of temperature over Africa characterized by high
5  values over the Sahara in summer as well as in spring (not shown), and low values in northern fall and winter (not shown). However moderate biases exist with cold biases over most of the area, and inter-model dispersion is also present, for instance summer temperatures in GFDL-ESM2M are about 2° C higher than temperatures in HadGEM2-ES or IPSL-CM5A-LR over western Sahara, and more over eastern Sahara. The bias-correction process improves quite well the simulations and provides



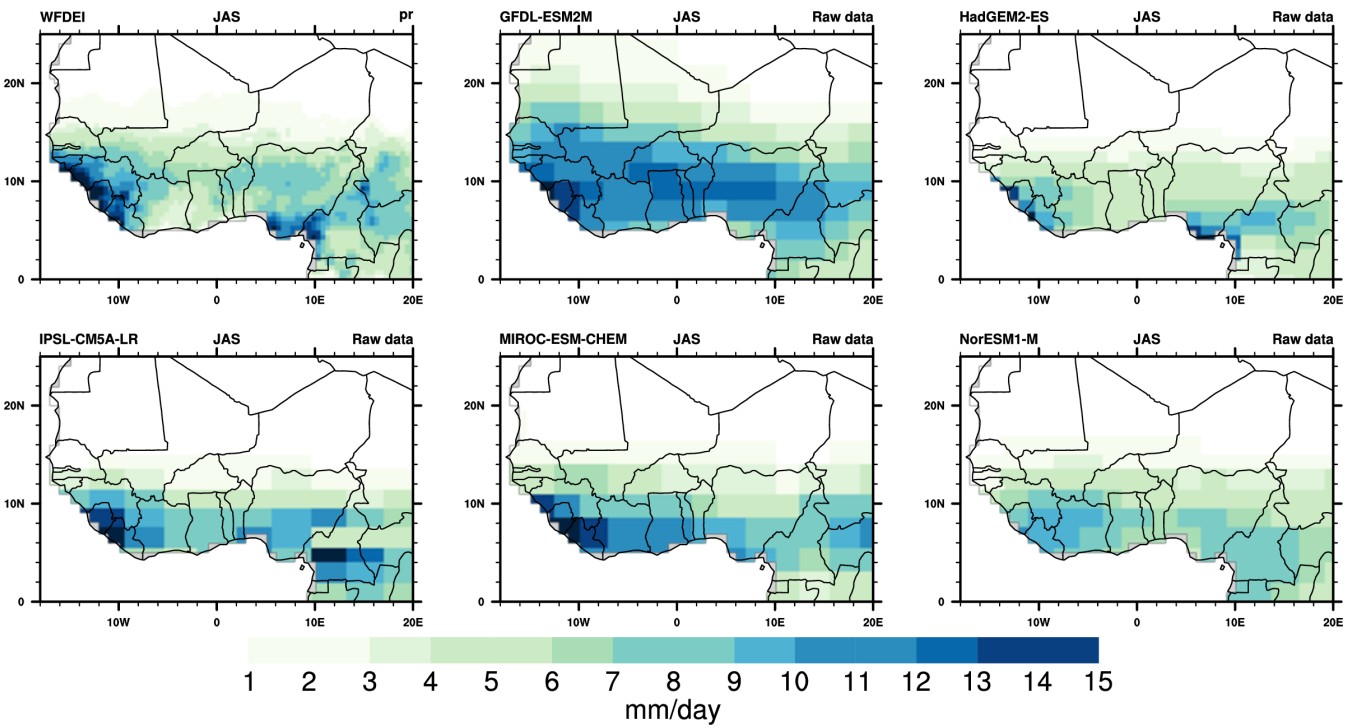

**Figure 4.** Same as Figure 2 but for prpitation rate in $\mathrm{mm\,day^{-1}}$

corrected mean seasonal fields very similar to WFDEI data, even at small spatial scales as for lower temperatures over Fouta-Jalon and Cameroon mountains (see Fig.S3). The Taylor diagrams (Fig.3) quantify this improvement very clearly for the 29 GCMs. The raw GCMs (Fig.3 left column) are quite scattered with spatial correlations with WFDEI distributed between +0.1 to more than +0.9. For the Sahel area, correlations are quite highin JAS (centred around +0.9) and the weakest in fall (around

+0.6; not shown), while for the Guinean area correlations are globally centred around +0.4. GCMs are also scattered in terms of normalized variances, from 0.6 to more than 2. The performance of the CDF-t bias-correction is clearly high since all the GCMs are very close to the WFDEI reference point. Taylor diagrams enable to compare the 5 GCMs used in ISIMIP in reference to WFDEI (Fig.3 middle column), for raw data, bias-corrected data by CDF-t and by ISIMIP method. WFD and EWEMBI data are also plotted. While CDF-t bias-corrected GCMs are very close to WFDEI, ISIMIP bias-corrected GCMs are centred around

WFD and near to WFDEI (correlation higher than +0.9 and normalized standard deviation close to 1). EWEMBI data is even more close to WFDEI with a correlation   +1 but with a normalized standard deviation that can be higher than for WFD. Fig.3 (right column) using EWEMBI data as the reference point, shows that bias-corrections provide again very good results with a better performance for CDF-t probably due to the higher similarity of WFDEI with EWEMBI than WFD. Same analyses have been carried out for tasmax and tasmin with similar results (not shown).

Fig.4, Fig.5 and Fig.S4 show similar results but for pr. The seasonal fields of WFDEI show the seasonal location of the ITCZ in JAS (Fig.4). Local maxima associated to highlands like Fouta Jalon or Cameroon mountains are also clearly





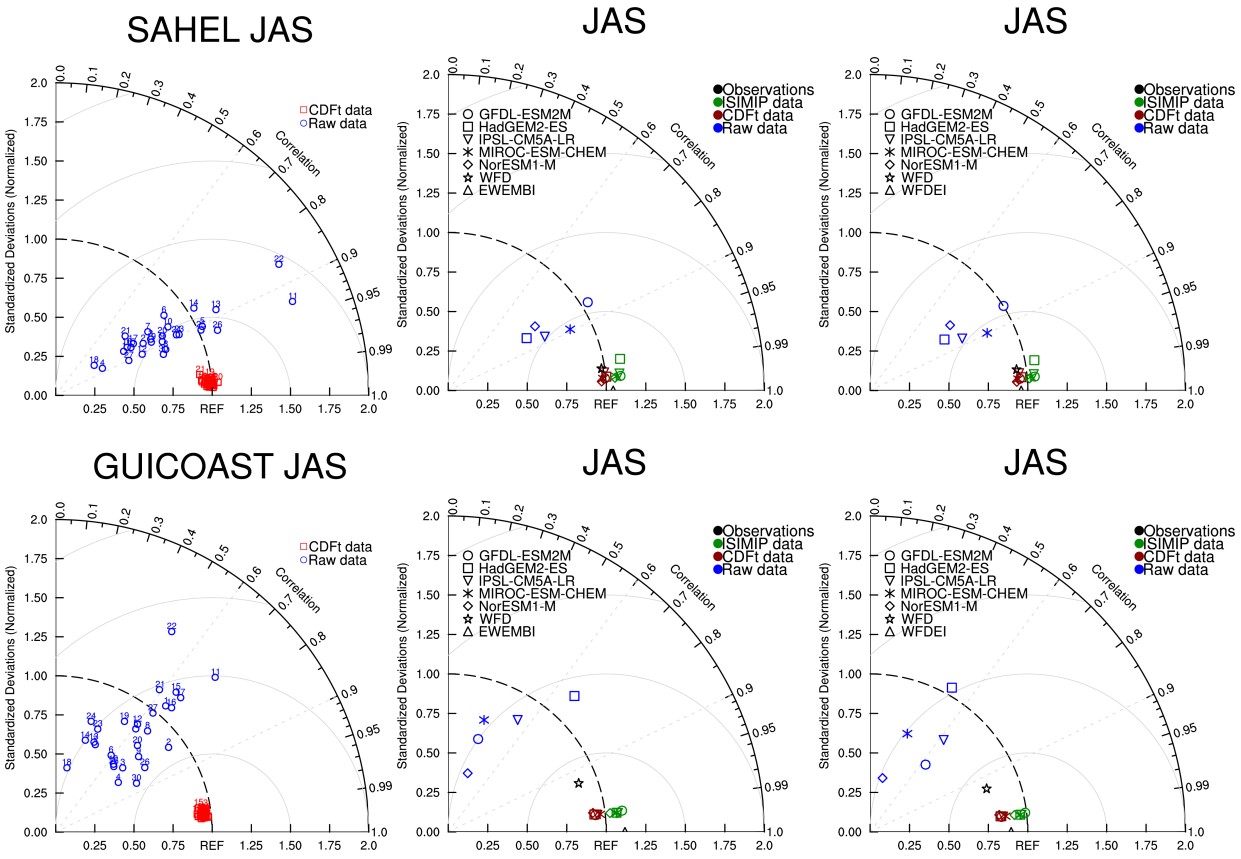

**Figure 5.** Same as Figure 3 but for prpitation rate

highlighted. Raw GCMs reproduce accurately this pattern but a lot of discrepancies can be noticed for all GCMs, both in terms of precipitation amplitude, spatial pattern and latitude extension. HadGEM2-ES has the weakest values while the four others produce precipitation amounts generally higher than WFDEI. The CDF-t bias-correction improves very efficiently the GCM mean seasonal precipitation fields since examination must be very detailed to discern differences with WFDEI fields and

5 between GCMs (see Fig.S4). This is clearly quantified with the Taylor diagrams over Sahel and Guinea areas in Fig.5. For raw GCMs the standardized variance is very scattered from 0.25 to more than 2. Spatial correlations are higher in Sahel (from +0.7 to +0.95) than in Guinea (from +0.2 to +0.8). The CDF-t bias-correction is quite effective in removing these biases and bringing the raw data closer to WFDEI, with some small remaining discrepancies, higher than for tas. The ISIMIP bias-correction is also effective due to the proximity between WFD and WFDEI. Finally the evaluation in respect to EWEMBI reference data is

10 good also, both CDF-t and ISIMIP bias-corrected data are nearby EWEMBI data.

Fig.6, Fig.7 and Fig.S5 show similar results, but for rsds. The mean seasonal fields of WFDEI show patterns with low values within the ITCZ area due to the high cloud coverage and high values on Sahara due to low moisture and cloud coverage. We have noticed previously (Fig.1) that differences exist between WFDEI, WFD and EWEMBI, with WFD rsds values as

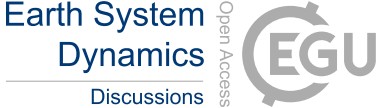

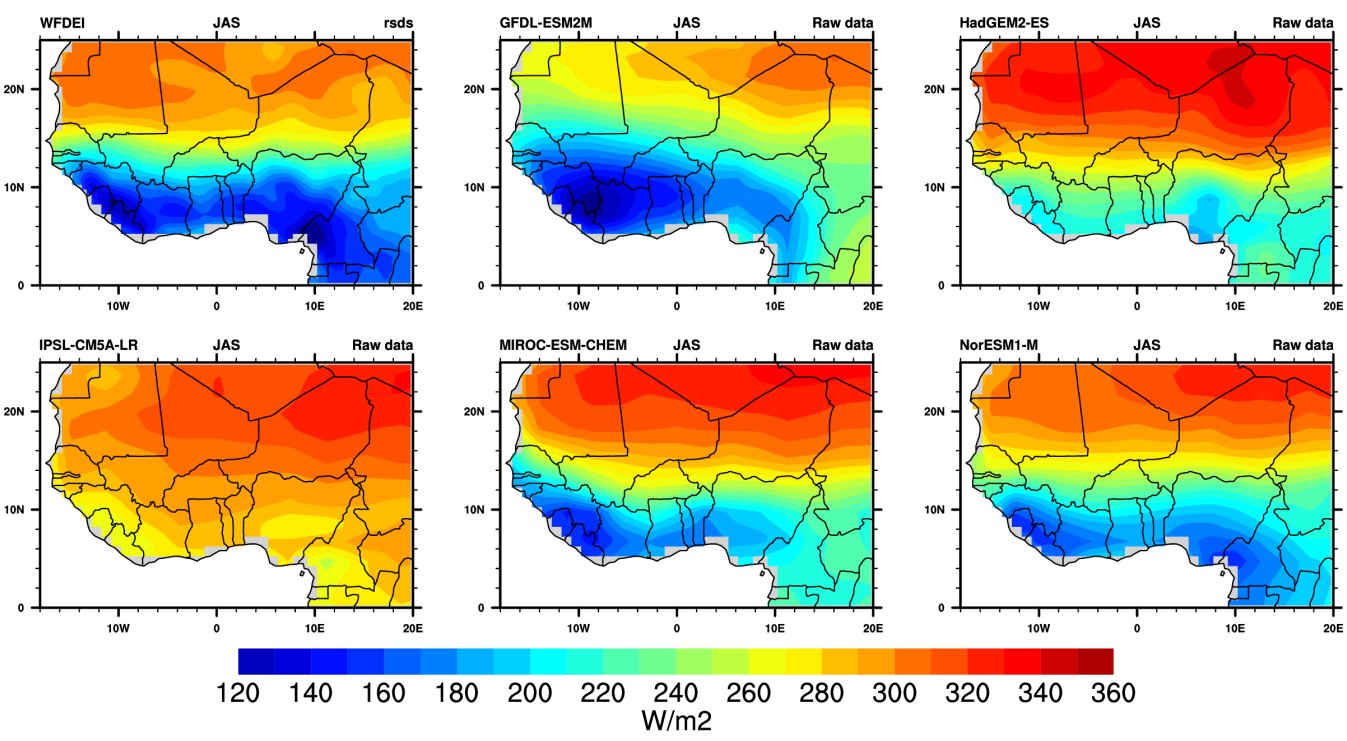

**Figure 6.** Same as Figure 2 but for solar radiation in $\mathrm{W\,m^{-2}}$

the weakest and EWEMBI values the highest in the ITCZ area, WFD values also the weakest and WFDEI values a bit higher than for EWEMBI over the Sahara. The five raw GCMs have, in agreement with their precipitation mean seasonal fields, a reasonable latitudinal evolution of low rsds values associated to the ITCZ but the range of rsds differences with WFDEI data as well as the inter-GCMs dispersion are very high. It shows an overall positive bias over the area. The CDF-t bias-correction is

once more very effective to remove biases respect to WFDEI data (see Fig.S5). The Taylor diagrams(Fig.7) provide some more quantification over the Sahel and Guinea areas. In terms of spatial correlation and normalized standard deviation in respect to WFDEI, raw GCMs have rather good performances over the Sahel (correlations higher than +0.8). Results are less good over the Guinea area (correlations less than +0.8) with a high dispersion. The ISIMIP bias-correction does not improve the performance of raw GCMs because WFD rsds data are quite far from rsds WFDEI data. rsds EWEMBI data are also far from

WFDEI. This last point is confirmed on the Taylor diagnostics in respect to EWEMBI reference data. Both bias-corrected data from CDF-t and ISIMIP method stay rather far from EWEMBI and do not improve significantly the performance of raw GCMs.

  Fig.8 to Fig.10 display other features of the mean fields in terms of Hovmoëller diagrams computed over West Africa (15° W-15° E). They show the mean fields of EWEMBI, WFDEI and WFD (first row), and the five GCMs used in ISIMIP (row

2 to 6) for raw data (first column), CDF-t corrected data (second column) and ISIMIP bias-correction method (third row). For tas (Fig.8), WFDEI, WFD and EWEMBI are very similar and highlight the set-up of the high temperatures area associated with



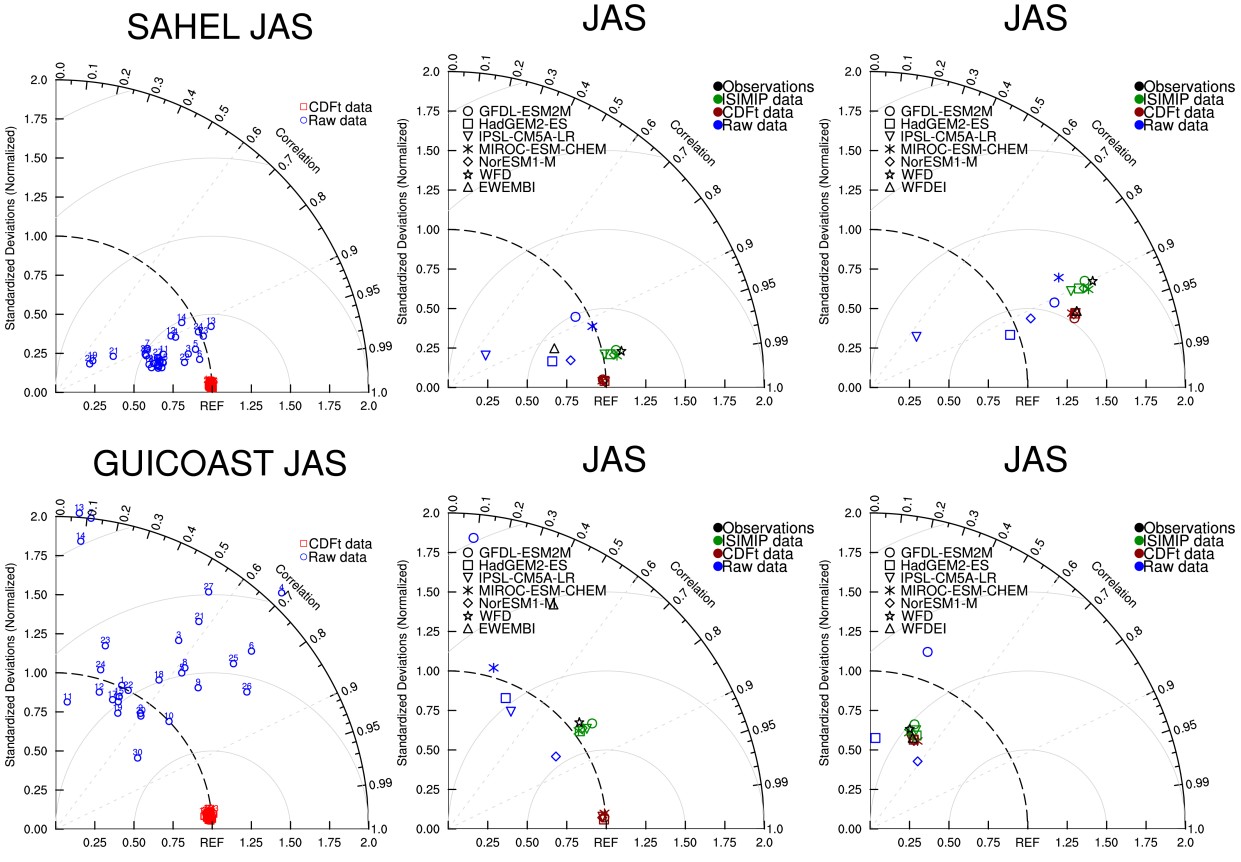

**Figure 7.** Same as Figure 3 but for solar radiation

the Saharan heat low in spring and summer (Lavaysse et al., 2009). Raw GCMs show a similar timing but their temperature values are lower by 2° C to 4° C depending on the model, and with an acceleration of the northward progression around June that is not present in observations. Bias-correction methods are effective to reduce these few discrepancies but few differences still remain as for instance a bit weaker temperature maximum around July in CDF-t corrected data.

5    Fig.9 shows similar diagrams but for pr. The Hovmoeller approach is a good way to depict the main characteristics of the ITCZ evolution over West Africa with a first rainy season during spring over the Guinea area followed by an "abrupt jump" to the north in June-July (Sultan and Janicot, 2003) and by a more progressive southward retreat at the end of the summer monsoon season leading in fall to a second weaker rainy season over the Guinean area. WFD and WFDEI are very similar with a bit noisier field for WFD, and EWEMBI shows more differences over the Guinean area, especially with a weaker second

10   rainy season. Raw GCMs have high discrepancies and produce mean fields quite different from a model to another one. In particular, precipitation data can be either very low (NorESM1-M) or very high (GFDL-ESM2M), and no GCM captures the abrupt northward shift of the ITCZ. The bias-correction methods (CDF-t using WFDEI, ISIMIP using WFD) are very effective in capturing back the main features of the ITCZ evolution, but differences still remain within the GCMs.





**Figure 8.** Hovmoëller diagrams of daily temperature (°C) averaged between 15° W and 15° E and for the period 1979-2001 for EWEMBI, WFDEI and WFD observations, each of the 5 GCMs for Raw data (1st column), CDF-t data (second column) and ISIMIP data (3rd column).

Fig.10 shows similar diagrams but for rsds. The seasonal evolution is in agreement with tas and pr fields and depicts high solar radiation values over the Sahara and weak values following the ITCZ latitudinal excursion but with a small southward lag (consistent with a higher cloud cover of mid-level clouds, (see Roehrig et al., 2013)). WFD shows an overall negative bias with respect to WFDEI and EWEMBI, and WFDEI has a higher meridional gradient than EWEMBI with lower minimum values

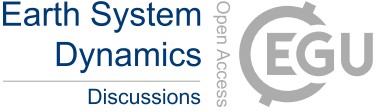

**Figure 9.** Same as Figure 8 but for precipitation rate in $\mathrm{mm\,day}^{-1}$

over the Guinea area and higher maximum values of the Sahara. The corrected GCM data are very close to their respective observation reference (WFD for ISIMIP, WFDEI for CDF-t) but still different between their two corrected versions due to the differences between WFD and WFDEI.



**Figure 10.** Same as Figure 8 but for solar radiation in W m$^{-2}$

## 4.3 Daily-based metrics over West Africa

In the following, similar diagnostics as in previous section are presented to evaluate other daily-based metrics. To reduce the number of figures in the core of the paper, some of them are presented in Supplementary Information (3 metrics in the core of the paper, 3 others in Supplementary Information).





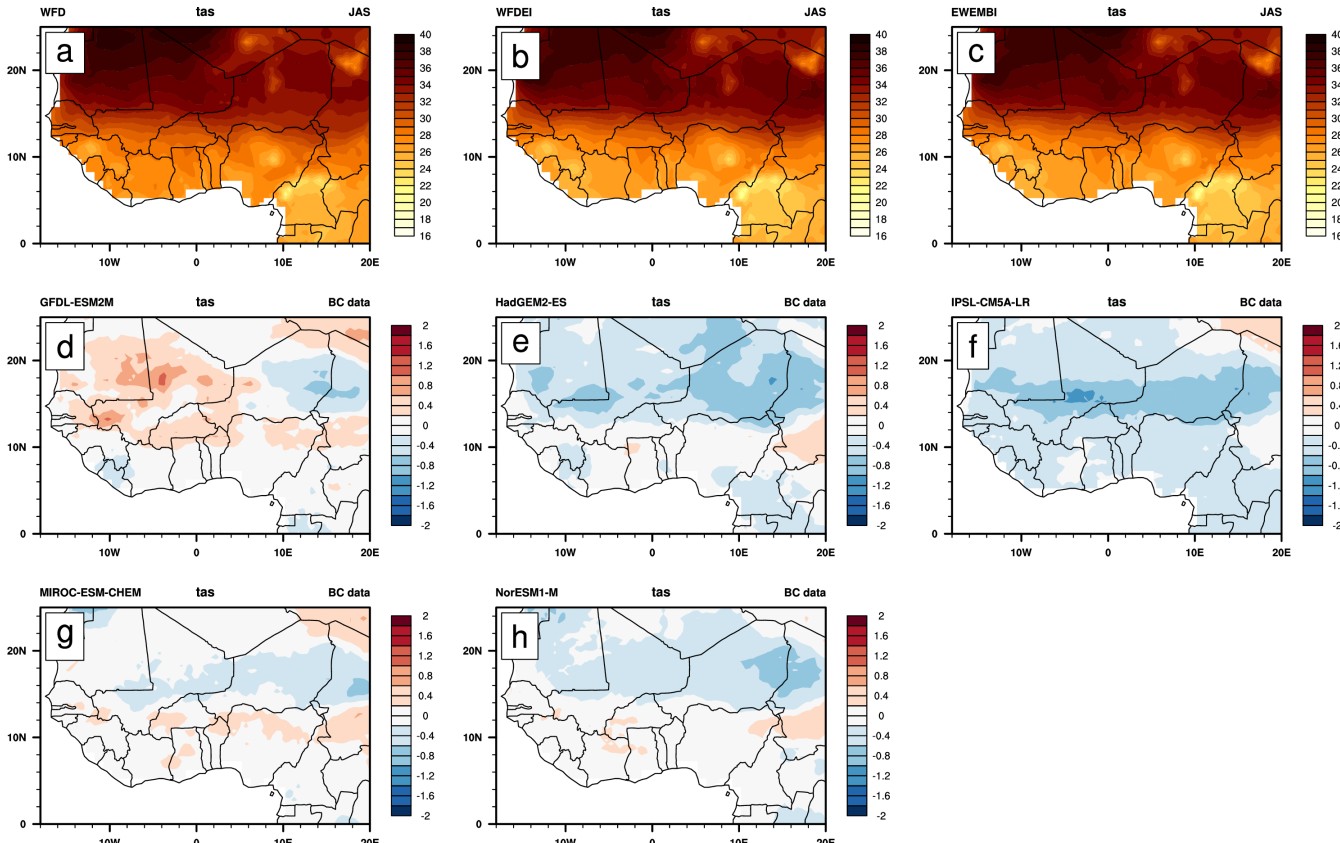

**Figure 11.** The 95$^{th}$ percentile of daily values for temperature from various observations dataset: WFD (a), WFDEI (b), EWEMBI (c) and difference relative to WFDEI data from 5 individual CDF-t bias-corrected models (d-h) over period 1979-2001

Fig.11 and Fig.12 address the tas 95$^{th}$ percentile of daily values. WFD, WFDEI and EWEMBI provide similar values in summer (Fig.11) with the highest values over Sahara in spring (up to 40° C, not shown), moving northward in summer, and with weaker values in fall (up to 32° C; not shown). WFD values appear as a bit higher than for the two other reference data sets. More to the south, in the Guinea area, the 95% percentile is between 30° C and 34° C. CDF-t bias-corrected data are also presented for the five GCMs used in ISIMIP. Some biases still remain but mostly lower than 1° C in absolute value. They are generally negative except for GFDL-ESM2M over the Sahara. Taylor diagrams depict the good performance of CDF-t bias-correction method. The highly scattered raw GCM data, especially over the Guinea area, move into a concentration zone very near WFDEI reference (Fig.12). ISIMIP bias-corrected data are also well concentrated near the WDF reference data but at some distance from WFDEI reference point. CDF-t method is also a bit better than ISIMIP one when one refers to EWEMBI reference data.

Fig.13 and Fig.14 provide similar analysis for the 95$^{th}$ percentile of daily pr. WFD, WFDEI and EWEMBI provide values consistent with the ITCZ location including high values over the mountain areas (Fig.13). WFDEI and EWEMBI have very



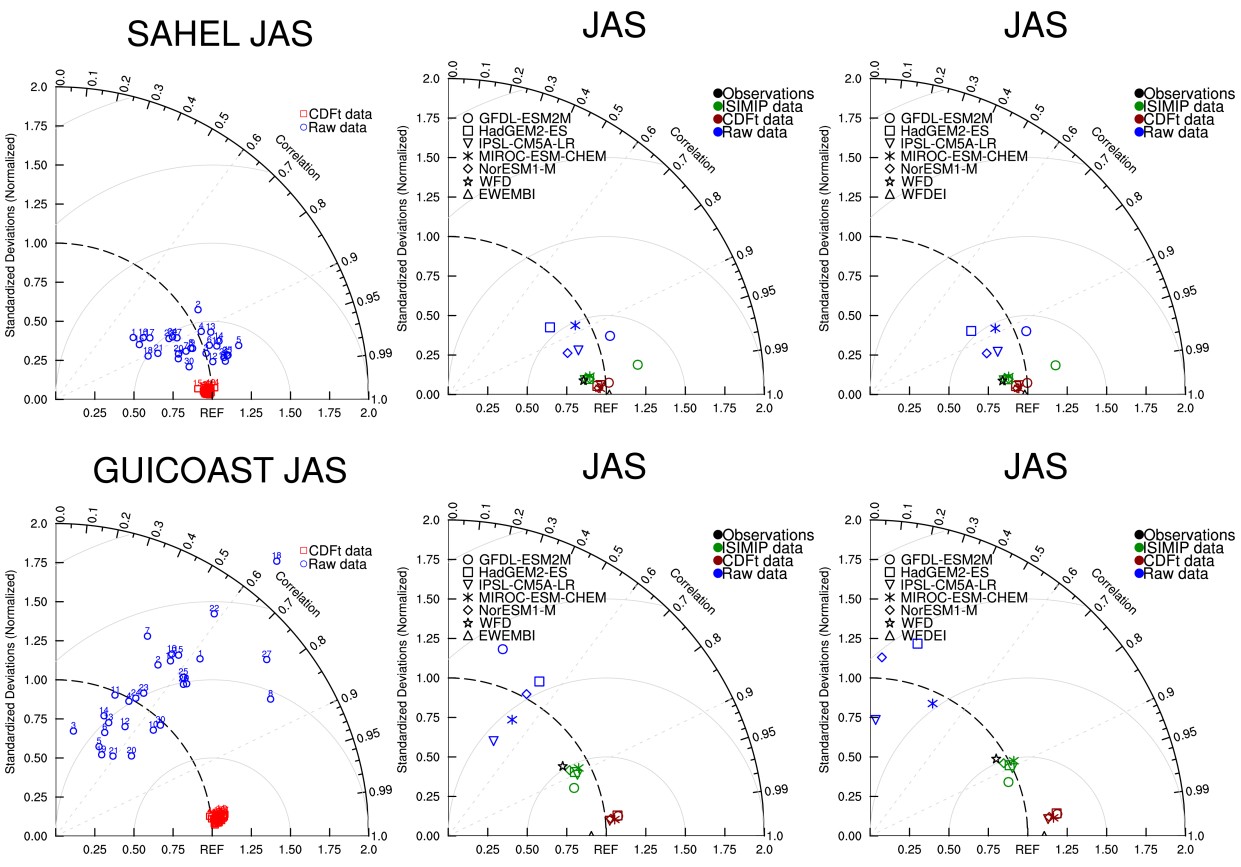

**Figure 12.** Same as Figure 3 but for the 95th percentile of near-surface temperature

similar fields while the range of values for WDF is very different, with values higher than $30\,\mathrm{mm\,day^{-1}}$ in the ITCZ in summer in contrast with values lower than $20\,\mathrm{mm\,day^{-1}}$ for the two other reference data sets. Similar range of differences is present over the Guinea area in springand to a lesser extent in fall (not shown). Such differences are also large over the mountain areas (Fouta Djalon, Cameroon). CDF-t bias-corrected GCMs have remaining weak biases in respect to WFDEI, lower than

5   $2\,\mathrm{mm\,day^{-1}}$, except for IPSL-CM5A-LR in summer with an extended area north of $10°$ N with values up to $+5\,\mathrm{mm\,day^{-1}}$. Taylor diagrams (Fig.14) show again the good performance of the CDF-t bias-correction method, the higher dispersion of the ISIMIP bias-corrected GCMs than CDF-t corrected GCMs in respect to their respective reference data set (WFD and WFDEI respectively), and the close distance of CDF-t bias-corrected data to EWEMBI reference.

      Fig.15 and Fig.16 provide similar analysis for the number of days with pr > $10\,\mathrm{mm\,day^{-1}}$. WFD, WFDEI and EWEMBI

10  provide values consistent with the ITCZ location including high values over the mountain areas (Fig.15). In opposite to the previous metric, WFD has a similar range of values that the two other reference data sets, with a small over-estimation. The spatial variance is higher than for the two previous metrics with a higher contrast between mountain and plain areas. Remaining biases in the CDF-t corrected data are localised over mountain areas with mostly negative biases, but also over plains in summer





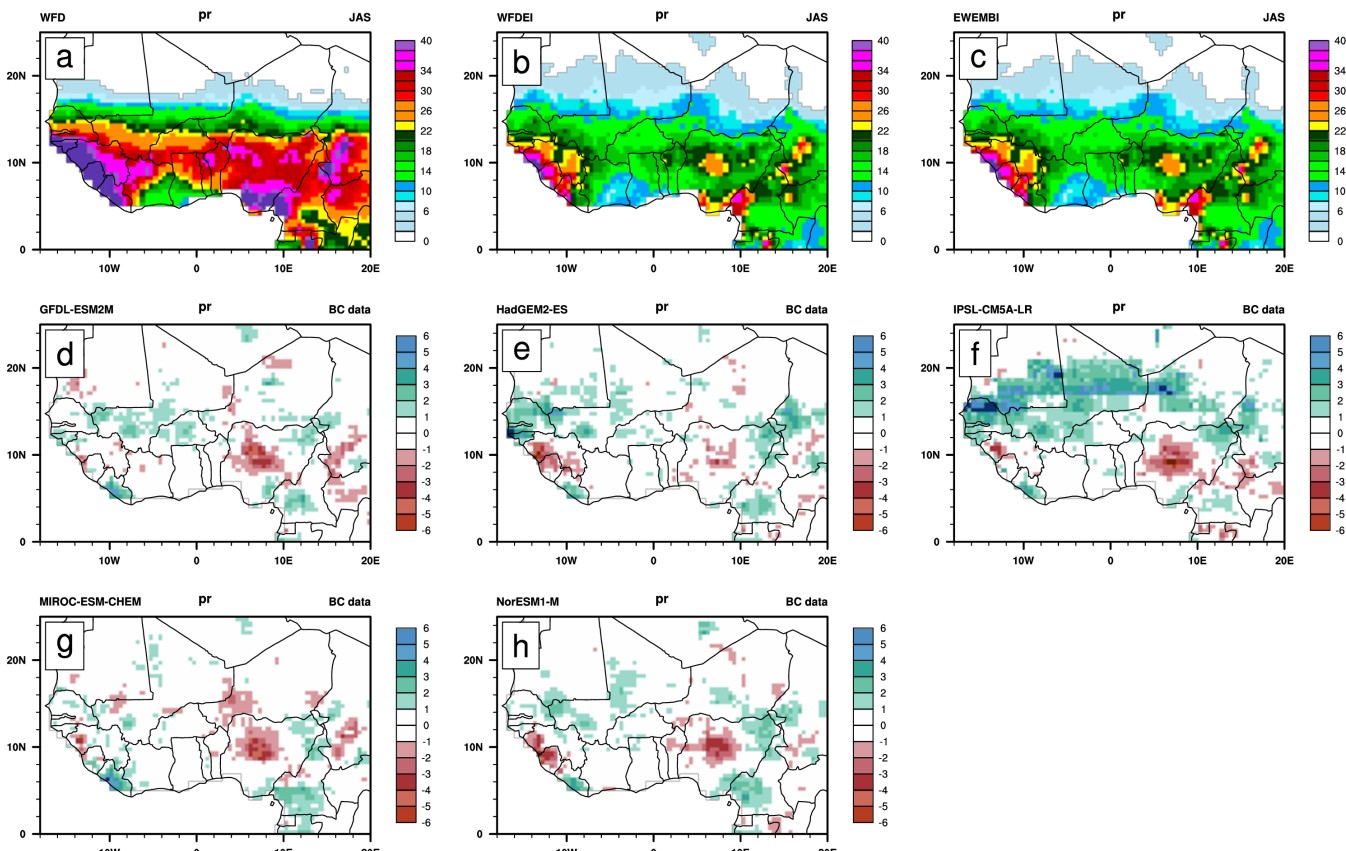

**Figure 13.** The 95$^{th}$ percentile of daily precipitation rate $(\mathrm{mm\,day^{-1}})$ from various observations dataset: WFD (a), WFDEI (b), EWEMBI (c) and difference relative to WFDEI data from 5 individual CDF-t bias-corrected models (d-h) over period 1979-2001

with mostly positive biases in the ITCZ area and especially extended in IPSL-CM5A-LR. Taylor diagrams (Fig.16) show once again a good performance of the CDF-t correction method. ISIMIP bias-corrected GCMs have a higher dispersion than CDF-t corrected GCMs in respect to their respective reference data set. CDF-t bias-corrected data are at a close distance to EWEMBI reference.

5 **4.4 Crop yields simulations and sensitivity to bias-corrected variables**

The question that we are now addressing is to evaluate what the sensitivity of simulated crop yields to raw and bias-corrected variables can be over West Africa. A crop model forced by atmospheric variables integrates biases and variability of these forcing datain a non-linear way. It is critical to evaluate if this integration may reduce or amplify the variability induced from these data.

10 This has been tested by using the crop model SARRA-O (System of Agroclimatological Regional Risk Analysis; version O). The model simulates yield attainable under water-limited conditions by simulating the soil water balance, potential and



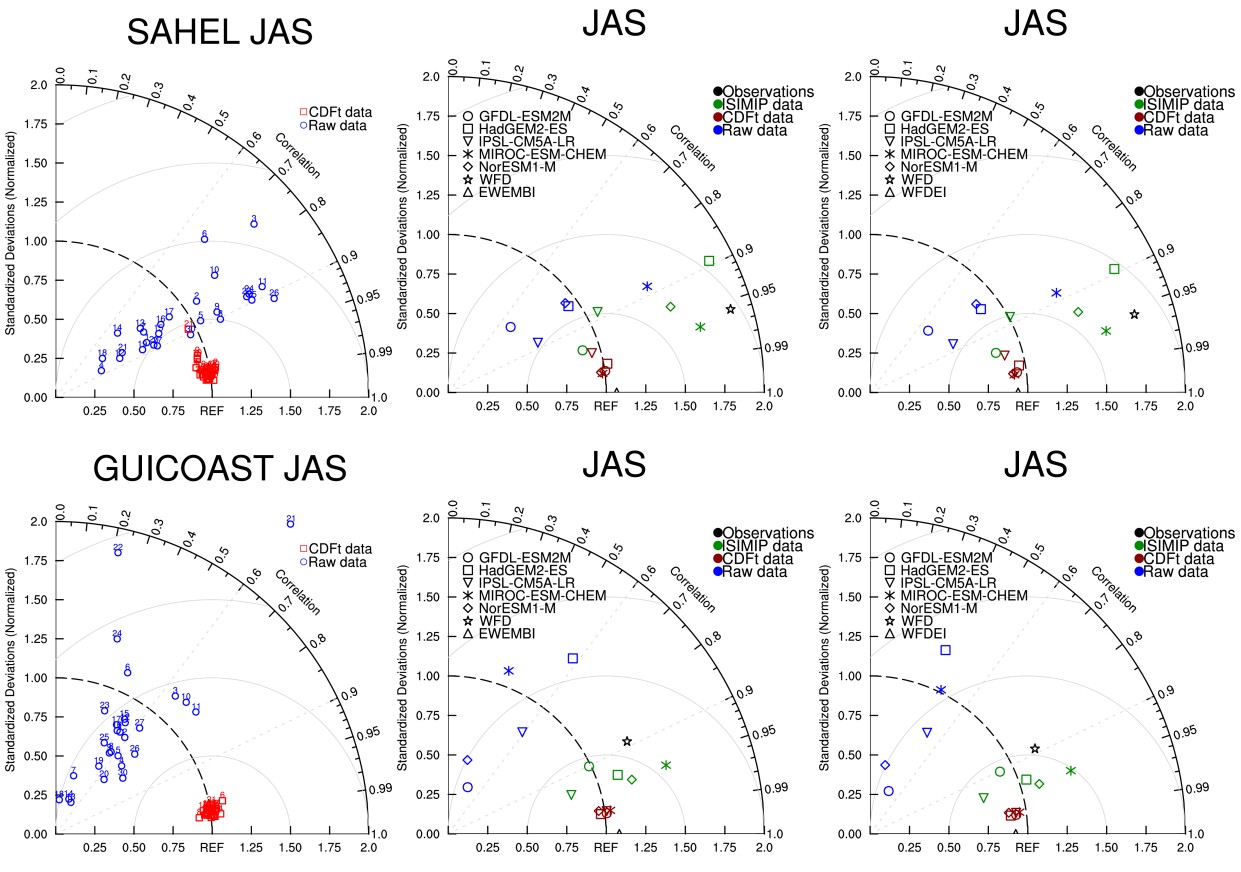

**Figure 14.** Same as Figure 3 but for the 95th percentile of daily precipitation rate ($\mathrm{mm\,day^{-1}}$)

actual evapotranspiration, phenology, potential and water-limited carbon assimilation, and biomass partitioning (see Kouressy et al., 2008, for adetailed review of model concepts). The simulation of these processes makes SARRA-O particularly suited for the analysis of climate impacts on cereal growth and yield in dry tropical environments (see for instance Sultan et al., 2013). Several sensitivity simulations have been carried out. First SARRA-O has been forced for each year from 1979 to 2001

5  by WFD, WFDEI and EWEMBI data. Second IPSL-CM5A-LR model has been used to force SARRA-O over the same years, with raw, CDF-t bias-corrected and ISIMIP bias-corrected data. The simulations have been compared to the "GDHY" data set (1981-2001) of 1.125° gridded maize yields estimated from a combination of global agricultural data sets, country yield statistics and satellite-derived net primary production (Iizumi et al., 2014). Note that SARRA-O provides potential yields that can be different from observed yields, so this comparison with the GDHY data set must be considered as indicative only.

10  Finally, sensitivity to individual variables has been conducted by comparing the SARRA-O simulation forced with WFDEI data with simulations where one WFDEI variable is replaced by the corresponding raw IPSL-CM5A-LR data.

Fig.17 compares the simulated crop yields over the Sahel and Guinea areas between WFD, WFDEI, EWEMBI, and raw, CDF-t and ISIMIP bias-corrected IPSL-CM5A-LR model. GDHY data are also shown as evaluation. Over the Guinea

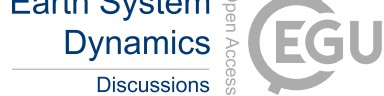

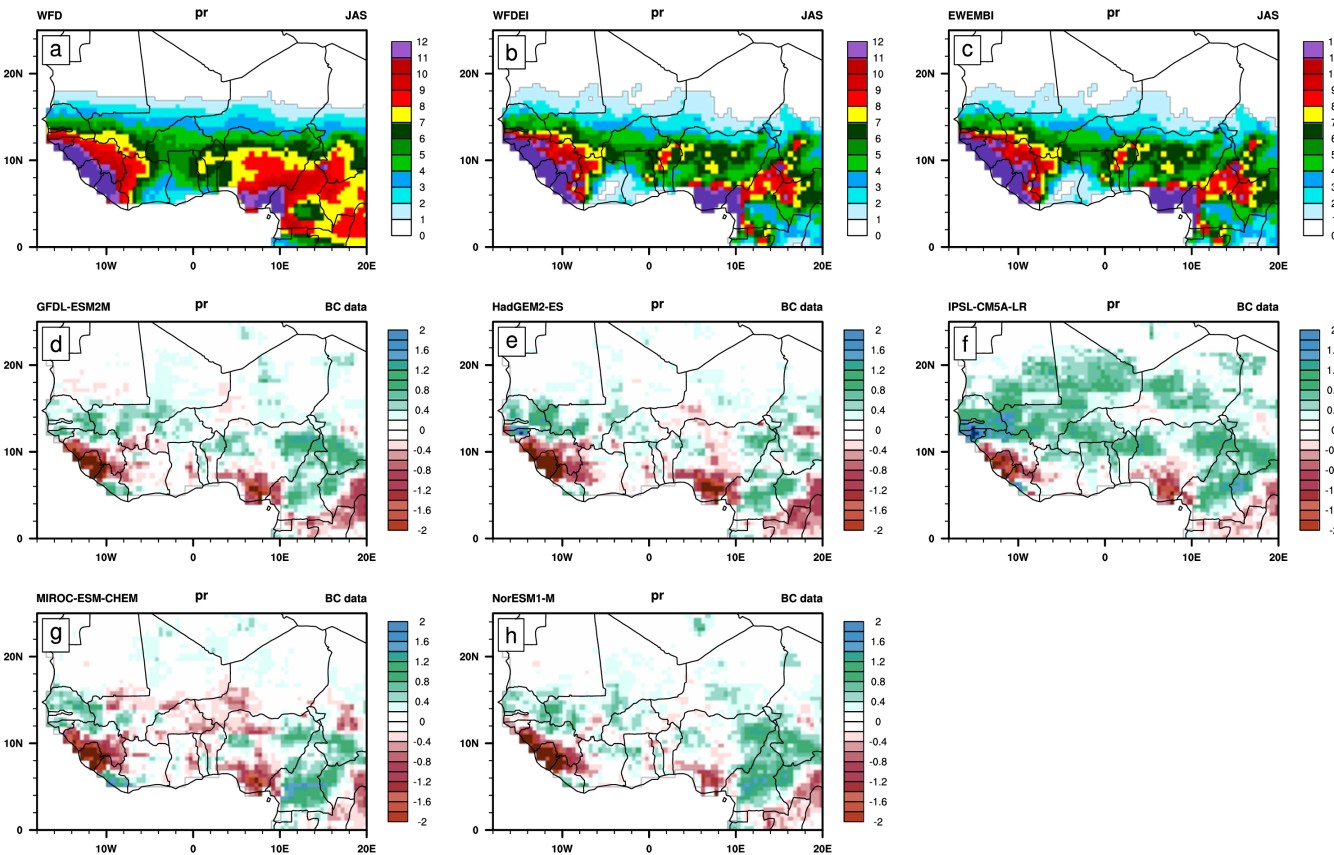

**Figure 15.** Seasonal mean of number of day with precipitation greater or equal to $10\,\mathrm{mm\,day^{-1}}$ from various observations dataset: WFD (a), WFDEI (b), EWEMBI (c) and difference relative to WFDEI data from 5 individual CDF-t bias-corrected models (d-h) over period 1979-2001

area, the differentiation of ensembles of simulations is quite clear. Raw IPSL-CM5A-LR simulation has the highest yields ( $2200\,\mathrm{kg\,ha^{-1}}$ ) while WFD and associated ISIMIP bias-corrected simulations have the lowest yields ( 240 and $180\,\mathrm{kg\,ha^{-1}}$ respectively). The four remaining simulations, based on WFDEI and associated CDF-t bias-corrected data, EWEMBI and GDHY data, have intermediate yields, between 700 and $1000\,\mathrm{kg\,ha^{-1}}$. So it is shown first that SARRA-O maize yields are

5  quite sensitive to the different forcing data sets, second that WFD leads to simulated yields far from the GDHY data while WFDEI and EWEMBI leads to quite better yields, and finally that raw GCM and GCM corrected with WFD are also quite far from validation data while GCM corrected with WFDEI has a rather good performance. The simulation forced by EWEMBI has a higher valuethan WFDEI ( 760 and $1030\,\mathrm{kg\,ha^{-1}}$ respectively) and GDHY data show yields ranging between WFDEI and EWEMBI ( $980\,\mathrm{kg\,ha^{-1}}$ ), close to EWEMBI. Over the Sahel area, the curves are closer but some similar conclusions can be

10 drawn. WFD and associated ISIMIP bias-corrected simulations provide the lowest yields ( 400 and $370\,\mathrm{kg\,ha^{-1}}$ respectively). WFDEI, EWEMBI and CDF-t bias-corrected simulations are very close ( 660, 650 and $710\,\mathrm{kg\,ha^{-1}}$ respectively). Finally in contrast to the Guinea area, GDHY data has the highest yields ( $980\,\mathrm{kg\,ha^{-1}}$ ), far from other simulations, and raw simulation



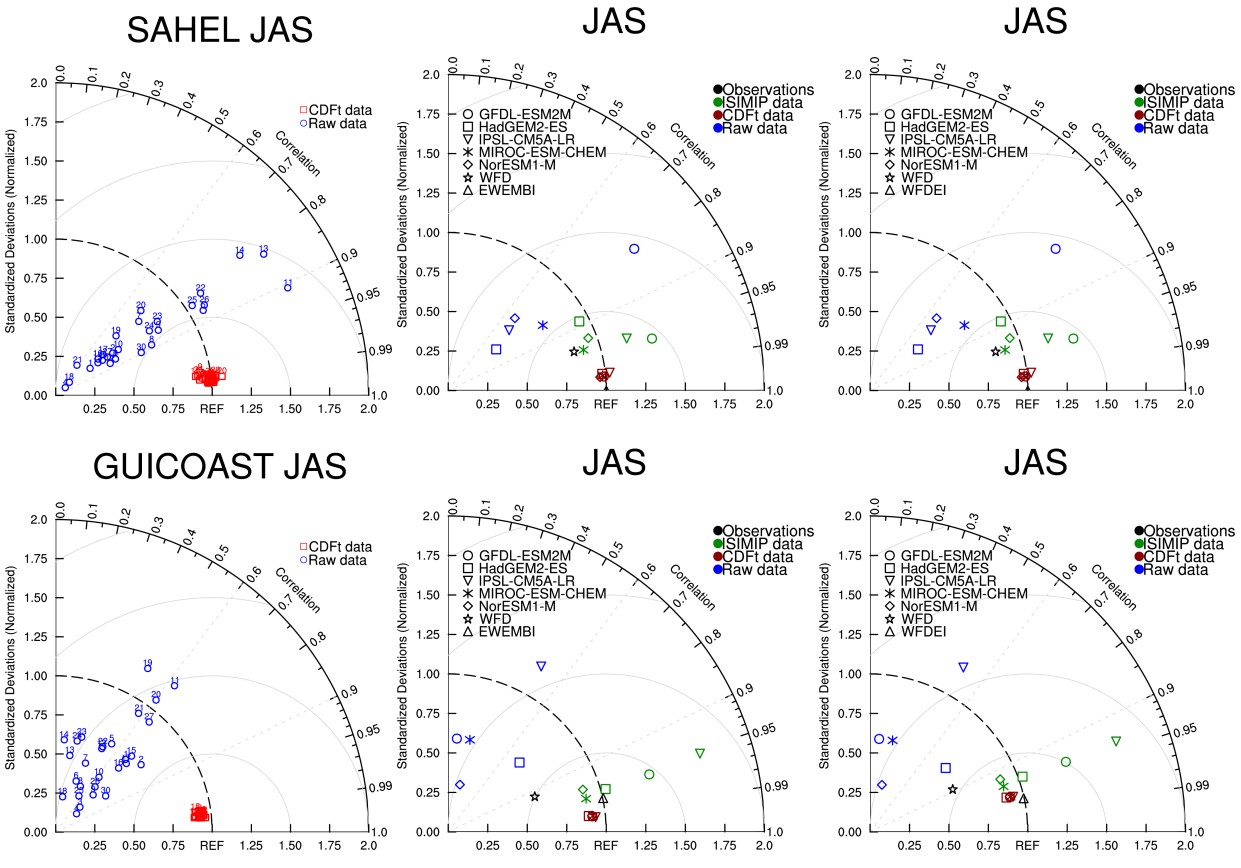

**Figure 16.** Same as Figure 3 but for number of day where precipitation is greater or equal to $10\,\mathrm{mm\,day^{-1}}$

( $590\,\mathrm{kg\,ha^{-1}}$ ) is close to WFDEI, EWEMBI and CDF-t bias-corrected simulations. This last point is quite surprising since raw IPSL-CM5A-LR data have large biases.

Fig.18 shows the maps of mean simulated yields for raw, WFDEI and CDF-t bias-corrected, WFD and ISIMIP bias-corrected, EWEMBI simulations, and GDHY data. For raw simulations, yields are highly underestimated over the central

5  Sahel but highly overestimated over the western Sahel and especially near the Fouta-Jalon. The boundary between the Sahel and Guinea boxes being at 10° N, the spatial average over the Sahel combine positive with negative biases in respect to WFDEI. Other maps show that yields obtained from EWEMBI are closer to GDHY data than yields from WFDEI, mostly due to better realistic values over the Guinea area (see also Table 2). Underestimation of yields simulated from WFDEI over Fouta-Jalon and over southern Cameroon and south-eastern Nigeria can be clearly associated with underestimation of WFDEI Rsds compared to

10  EWEMBI rsds (see Fig.1). Comparisons on WFDEI and EWEMBI interannual time series of yields and associated tas, pr and rsds on individual grid points in these areas confirm that these yields differences are linked exclusively to rsds differences while tas and pr are similar (not shown). Finally maps of simulated yields from WFD and ISIMIP bias-correction ISIMIP confirm





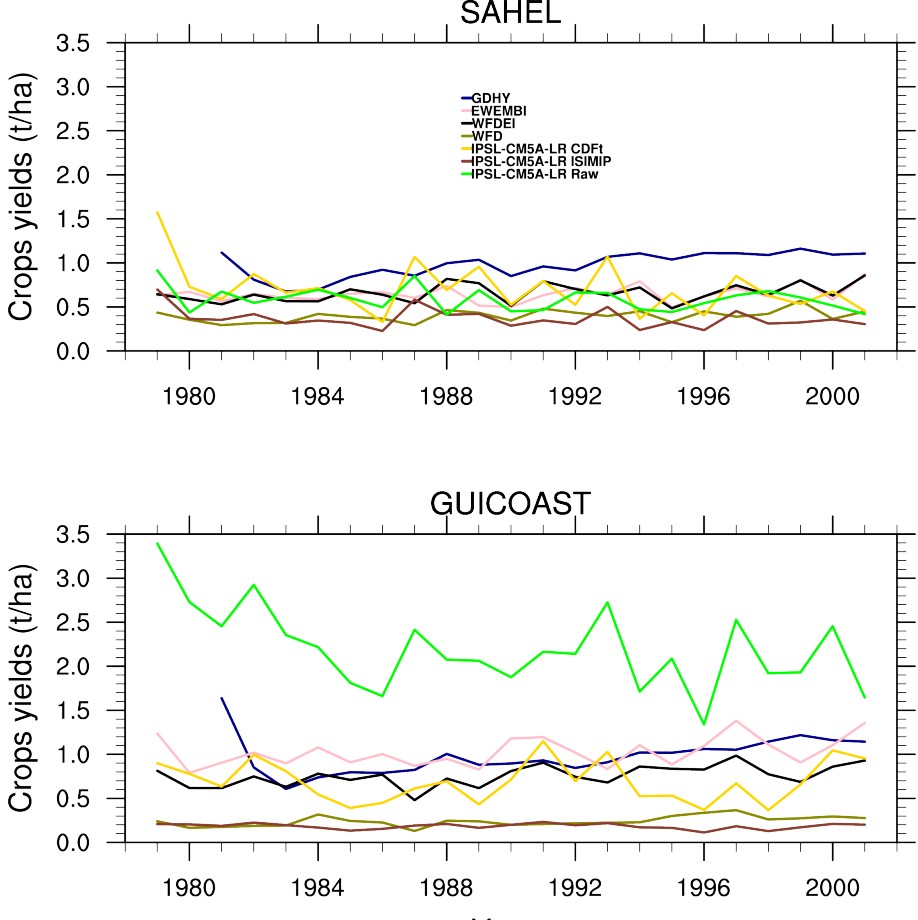

**Figure 17.** Time series of crop maize yield over the Sahel forced by IPSL-raw, IPSL-CDFt, IPSL-ISIMIP, and WFD, WFDEI, EWEMBI over 1979-2001

the highly weak values over the whole West Africa, with as for WFDEI and CDF-t bias-correction, an under-estimation due to rsds south of 10° N (not shown).

To go further, a sensitivity analysis to individual variables has been conducted by comparing the SARRA-O simulation forced with WFDEI data with simulations where one of these WFDEI variables is replaced by the corresponding raw IPSL-CM5A-LR data. These variables are pr, rsds, tasmin and tasmax, and also rsds from ISIMIP bias-corrected IPSL-CM5A-LR (using WFD as reference). Table 2 shows the results and the resulting biases in respect to WFDEI simulation, obtained for the Sahel and Guinea areas in reference to the previous simulations as well as all new WFDEI simulations. Biases are very weak with tasmin-tasmax simulations (WFDEItminmax), a bit higher for pr simulations (WFDEIpr), a bit higher for rsds, (WFDEIrsds) and drastically large for rsds from ISIMIP bias-corrected simulations (WFDEIWFDrsds; 340 and 560 in respect to  660 and 760 $\mathrm{kg\,ha^{-1}}$ for the Sahel and Guinea areas respectively). So rsds appears as a very critical variable for maize



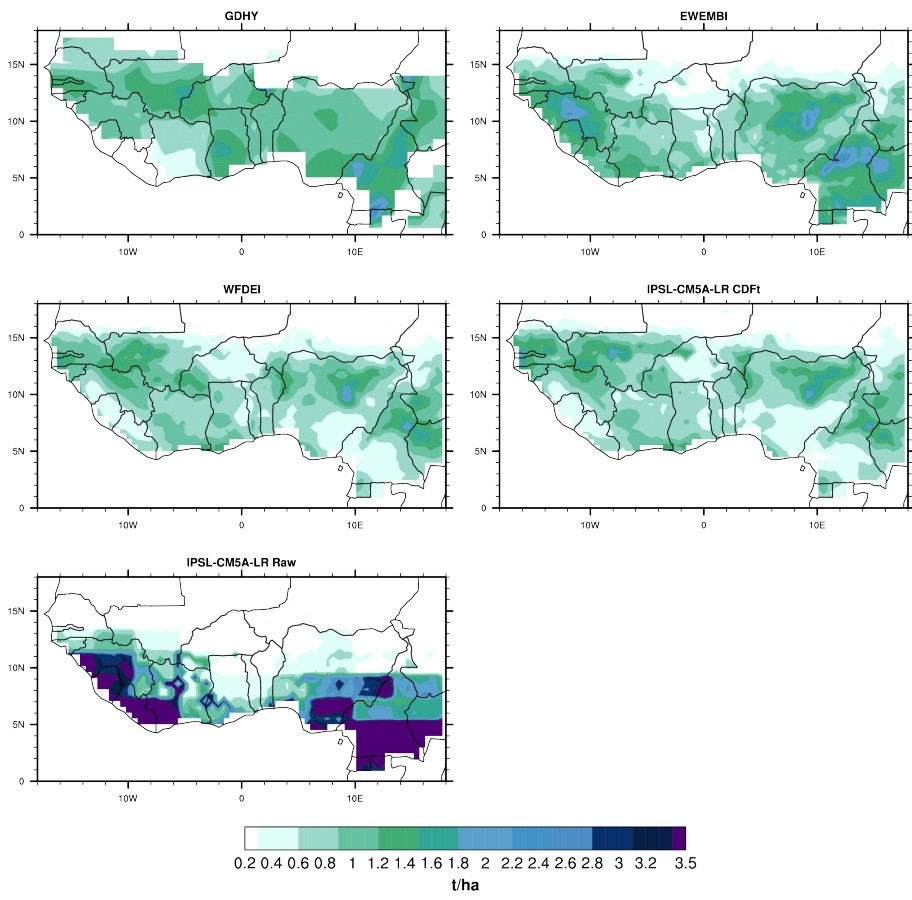

**Figure 18.** Temporal mean of maize yield for IPSL-raw, IPSL-CDF-t, IPSL-ISIMIP, WFDEI, WFD, and GDHY over 1979-2001

yields simulated with SARRA-O, confirming a previous study based on an older version SARRA-H of crop model (Oettli et al., 2011). It indicates also that WFD data and related bias-corrected simulations should not be used anymore.

Fig.19 shows the projection of maize yields from SARRA-O forced by ISPL-CM5A-LR in terms of raw, CDF-t bias-corrected and ISIMIP bias-corrected data, on one hand from 1950 to 2099 over the Sahel and Guinea boxes, and on the other hand as the map of yield from CDF-t bias-corrected data on 1979-2001 and 2077-2099. In agreement with the previous analysis, ISIMIP bias-corrected forcing data (with WFD as reference data) lead to the lowest yields both over the Sahel and Guinea areas on the present time but also over the whole 21[st] century. Over Guinea area, the very high simulated yields coming from raw data are drastically reduced with CDF-t bias-corrected forcing data (with WFDEI as reference data) while over Sahel area these yields are rather similar. After CDF-t bias-correction, yields are quite similar over the two areas. In agreement with the previous analysis, ISIMIP bias-corrected forcing data (with WFD as reference data) lead to the lowest yields both over the Sahel and Guinea areas on the present time but also over the whole 21[st] century. Interannual variability of simulated yields is



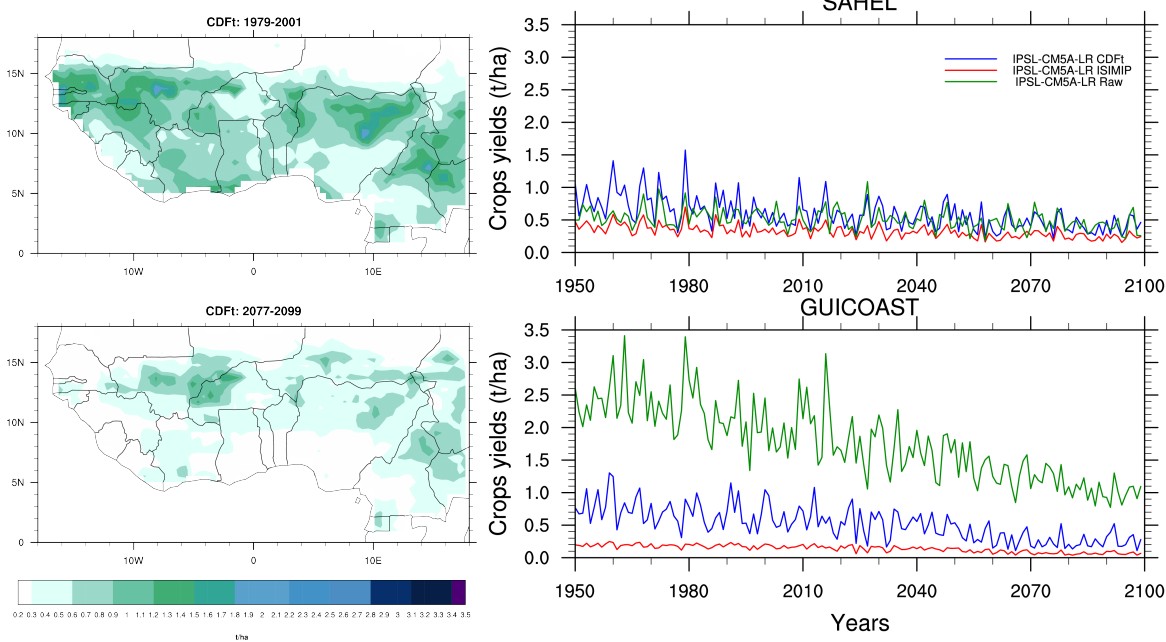

**Figure 19.** Projections in crops yields over Sahelian box and Guinean box forced by IPSL-CM5A-LR raw data (green line), BC data with CDF-t (blue line), ISIMIP BC data (red line).

**Table 2.** Bias (in $\mathrm{kg\,ha^{-1}}$) sensitivity experiments (see details in the text).

|  | SAHEL | | GUICOAST | |
|---|---|---|---|---|
|  | **Mean** | **Bias** | **Mean** | **Bias** |
| WFDEI | 658 | 0 | 757 | 0 |
| WFD | 398 | -260 | 241 | -516 |
| EWEMBI | 646 | -12 | 1029 | 272 |
| GDHY | 979 | 321 | 978 | 221 |
| IPSL-CM5A-LR Raw | 586 | -72 | 2201 | 1444 |
| IPSL-CM5A-LR CDFt | 706 | 48 | 693 | -64 |
| IPSL-CM5A-LR ISIMIP | 367 | -291 | 184 | -573 |
| WFDEIpr | 668 | 10 | 716 | -41 |
| WFDEIrsds | 717 | 59 | 786 | 29 |
| WFDEItminmax | 658 | 0 | 767 | 10 |
| WFDEIWFDrsds | 317 | -341 | 195 | -562 |



proportional to the mean with a very weak variability for ISIMIP yield and higher variability for CDF-t and raw simulations. All projections show a clear decrease of maize yields by a factor of 2 over all of West Africa along the 21$^{st}$ century.

## 5  Conclusions

The objective of this paper is to present a new data set of bias-corrected CMIP5 GCMs daily data over Africa using
CDF-t method, a method that has widely proved its efficiency but was not yet applied on Africa. It has been applied over the period 1950-2099 combining historical runs and RCP scenarios with 29/27/20 GCMs for RCP8.5/4.5/2.6 respectively. These bias-corrections have been applied to six variables critical for agricultural impacts, precipitation, mean near-surface air temperature, near-surface maximum air temperature, near-surface minimum air temperature, surface down-welling shortwave radiation, and wind speed.

The use of different bias-correction methods based also on different reference data sets contributes to the total uncertainty in climate projections and can contribute in some contexts more than the use of different GCMs or RCMs (Iizumi et al., 2017). So using multiple bias-correction technics and reference data sets is highly recommended. In this context, CDF-t bias-corrected GCM data have been compared to the 5 GCMs ISIMIP bias-corrected data, and the impact of the different reference data sets, WFD (used in ISIMIP bias-corrections), WFDEI (used in CDF-t bias-corrections) and the more recent EWEMBI (used in the
second version of ISIMIP bias-corrections), has been examined in details. Crop simulations have been also carried out to test how the impact of bias-corrections in forcing data (temperature, precipitation, surface down-welling shortwave radiation) is integrated in terms of crop (maize) yields. Finally bias-corrections have also been presented in the context of RCP8.5 scenarios.

The whole observational period, 1979-2013, has been chosen to calibrate the bias-correction process. It has been shown that it provides a more robust correction without leading to over-estimation of the fit between observations and bias-corrected
data, and that the impact of using various calibration sub-periods on the time evolution over the 21$^{st}$ century is weak.

The evaluation of CDF-t bias-correction applied to the 29 GCMs, both on mean seasonal data and on daily-based metrics, has shown that CDF-t is very effective in removing the biases in respect to the reference WFDEI data and reducing the high inter-GCMs scattering. It has also shown some distance, depending on variables and metrics, with bias-corrected ISIMIP GCM data, mainly due to the differences between WFDEI and WFD reference data. WFDEI (and associated CDF-t bias-corrected
GCMs) appears closer to EWEMBI than WFD (and associated ISIMIP bias-corrected GCMs). Metrics based on temperature are very close for the three reference data sets and some differences exist in precipitation-based metrics. In contrast, significant differences have been highlighted in terms of surface down-welling shortwave radiation. This has some consequences in terms of crop (maize) yields over West Africa. Sensitivity simulations on one GCM have shown that bias-corrections improve the yields simulated by the raw GCM, but that ISIMIP bias-corrected GCM still underestimate them, as CDF-t bias-corrected GCM
do, while being closer to observed yield estimates. EWEMBI provides the closest yields to observed estimates. This is mainly due to surface down-welling shortwave radiation whose values are under-estimated in WFDEI south of 10° N. Finally, in agreement with maize yield sensitivity simulations, projections of future yields over West Africa have quite different levels depending on bias-correction method. However they show all a similar relative decreasing trend over the 21$^{st}$ century.



The main perspective of this work is to go on exploring the uncertainty linked to bias-correction methods and their associated reference data in RCP climate scenarios by producing a second version of this bias-corrected 29 GCMs ensemble over Africa using more recent reference data like EWEMBI or others as those used in AgMIP based on other reanalyses (AgMERRA or AgCFSR, (Ruane et al., 2015)). The main divergence between all those reference data sets are probably

expected from surface down-welling shortwave radiation. Bias-correction for other variables useful for user-based metrics as specific humidity is also scheduled. Comparison between CDF-t and ISIMIP bias-corrections methods based on the same reference data set is also on-going.

The CFD-t bias-correction has been applied independently for each of the six variables. However this may be a problem since existing spatial coherency and dependence among variables maybe destroyed by the application of univariate calibrations.

Recently, to address this issue, improved calibrations have been developed in terms of multivariate correction, spatial and/or temporal dependences (see for instance Vrac and Friederichs, 2015, for a synthesis). Implementation of more sophisticated methods using multivariate correction is also on-going.

This work constitutes a first step in producing bias-corrected data sets over Africa within AMMA-2050. Evaluation of the results has been also carried out, especially over West Africa, on a list of priority users-based metrics that was discussed and

selected with stakeholders. An atlas is in preparation that will provide extensive results over Africa to the FCFA stakeholders and end-users communities. These communities will be accompanied by FCFA climate scientists,in order to be aware of the way to use these data and their limitations.

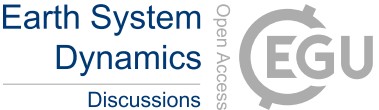

*Acknowledgements.* The research leading to these results has received partial funding from the NERC/DFID Future Climate For Africa programme under the AMMA-2050 project, grant number NE/M019934/1. The lead author has been also supported by IRD. M. V. has been partly funded by the ANR StaRMIP project. We acknowledge the World Climate Research Programme's Working Group on Coupled Modelling, which is responsible for CMIP, and we thank the climate modelling groups (listed in Table1 of this paper) for producing and

5    making available their model output. For CMIP the U.S. Department of Energy's Program for Climate Model Diagnosis and Intercomparison provides coordinating support and led development of software infrastructure in partnership with the Global Organization for Earth System Science Portals. The authors thank also EU Watch project and its members for data availability and ISIMIP data (doi:10.5880/PIK.2016.001 for ISIMIP Fast Track and http://doi.org/10.5880/pik.2016.004 for EWEMBI dataset). The CDF-t bias-corrected CMIP5 data over Africa is currently under embargo. Upon the expiration of the embargo the data will be made available by the AMMA-2050 project.





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
