# Peer review of "A bias-corrected CMIP5 dataset for Africa using CDF-t method. A contribution to agricultural impact studies."

_Earth System Dynamics, 2017_

## Referee Comment (RC1) · T. Iizumi (Referee) · 4 Jan 2018

General comments:

The presented study developed the bias-corrected CMIP5 GCM daily dataset using a combination of CDF-t method and WFDEI (and EWEMBI) and then compare with the ISI-MIP dataset which based on different bias-correction method and reference data (WFD). Some extreme climate indices (daily-based metrics) as well as maize yield simulated by a crop model were compared across different datasets to characterize the quality of the bias-corrected GCM daily dataset provided in this study. I respect the authors' efforts conducting this comprehensive analysis. Although this study analyzes

only West Africa, their findings have implications especially for agricultural impact studies across the world. I only have a few concerns (listed below) and believe that most of them are minor. I recommend the acceptance of this manuscript with minor revision.

Specific comments:

1. Section 3.1. Although the detailed description of CDF-t method may be available in earlier study (Michelangeli et al. 2009, Vrac et al., 2012, 2016), a more completed explanation of key characteristics of the method in this section is unavoidable to make this manuscript stand-alone. Otherwise readers have to scratch around for. Particularly, it would be great if you could add an brief explanation whether the method forces the maximum (or minimum) value of a climatic variable in the future projection to be the same with that in historical period or not.

2. Section 3.3. I am not convinced whether the current design of the sensitivity analysis is appropriate to evaluate the sensitivity of correction to the length of calibration period. When the data in 1979-1996 are used as the calibration subset, those in 1997-2013 are used as the validation subset; this is fine. But, there is no independent validation subset when the data in 1979-2013 are used as the calibration subset. And it is easily expected that biases in bias-corrected data become the smallest when all available data are used as the calibration data. Therefore, the conclusion that the correction with the longest calibration period leads to the smallest biases is not examined using independent data. However, I think this part is not essential in this study. Removing or reanalysis are possible for this part.

3. Fig. 5. In GUICOAST JAS, ISI-MIP bias-corrected precipitation data distantly distributed from WFD. Why? This is unreasonable because most methods including the ISI-MIP bias-correction method forces GCM data in the historical period similar to the reference data (WFD). A plausible explanation is necessary.

4. Fig. 14. Two tendencies are observed in this figure. One is that the 95th percentile precipitation values from ISI-MIP data largely varied by GCM compared to the spread
across the GCMs found in CDF-t method. Why? Another one is that in SAHEL JAS the IPSL-CM5A-LR data corrected by CDF-t method are relatively far from other GCMs of CDF-t data. A brief explanation is required.

5. P25L2. "It indicates also that WFD data and related bias-corrected simulations should not be used anymore." I think this is overstated. Please consider rephrasing or removal. I agree that a use of WFD leads to biased crop yields in crop model simulation mainly due to biases in solar radiation, as demonstrated in your analysis. However, simulated yield variability and/or projected future change in yields would not be affected in relative term when biases in solar radiation are a main issue (for instance, see Iizumi et al., 2010). Reliable projection of yield change in absolute term is challenging, and therefore projected relative change in yield is still only a main source of information for adaptation planning and other application.

6. Section 4.3. The presentation of daily-based metrics is relatively not well organized in the current manuscript. The analysis and findings on the daily-based metrics themselves are useful, but not comprehensive compared to earlier study examining daily-based metrics (e.g., Iizumi et al., 2017, JGR, doi:10.1002/2017JD026613). Why did you select a limited number of metrics for this analysis? More importantly, it seems that the importance of the analysis results is not equal to that of the analysis of crop model. The simulated maize yield is used as a metric in the current manuscript, as the daily-based metrics are did so; though the maize yield has much importance in the manuscript compared to the daily-based metrics. A justification to present a limited number of daily-based metrics is necessary.

Technical corrections

7. P1L2-3. Why has CDF-t method never been applied to Africa? Is that due to low availability of daily weather observations in that region? 8. P2L6. "robust" biases. Do you mean "persistent" biases? 9. P2L19. "high and robust biases". Do you mean "large and persistent biases"? 10. P3L24. "northern summer". Is this "northern hemispheric

summer" or "boreal summer"? 11. P'L7. "anomalies". Probably, "differences" is more appropriate here. 12. Fig. 3. "observations". This should read "references". 13. P12L6. "standardized variance". This should read "standardized standard deviation" by definition of Taylor diagram. 14. P21L6. "GDHY" are a hybrid of FAO country yield data, satellite-derived crop-specific vegetation index and global crop datasets on crop calendar, harvested area and production shares achieved by different growing season. Subnational yield statistics are used to validate the grid-cell yield estimates, but not used as the input to estimate grid-cell yields. 15. P23L9. "Rsds" -> "rsds"? 16. Fig. 19. The label of y-axis should be replaced by "maize yield" instead of "crop yield" to be more precise. 17. Table 2. The caption of the table needs to include sufficient information to interpret the results presented in the table. Saying "see detail in the text" is not acceptable from the viewpoint of readability.

References Iizumi, T., M. Nishimori, and M. Yokozawa, 2010: Diagnostics of Climate Model Biases in Summer Temperature and Warm-Season Insolation for the Simulation of Regional Paddy Rice Yield in Japan. J. Appl. Meteor. Climatol., 49, 574–591, https://doi.org/10.1175/2009JAMC2225.1

---

## Referee Comment (RC2) · Anonymous Referee #2 · 23 Jan 2018

**General comments**

The authors present a dataset of CMIP5 GCM output of daily accumulated precipitation, daily mean, minimum and maximum near-surface air temperature, daily mean surface downwelling shortwave and daily mean wind speed bias-corrected using the CDF-t method and the global gridded observational dataset WFDEI over Western Africa. They compare the raw and the bias-corrected data to WFDEI and 2 other observational datasets, namely WFD and EWEMBI. Unsurprisingly, the bias-corrected data feature much smaller biases relative to WFDEI than the raw data. Specifically, the authors look into biases of seasonal mean values and mean annual cycles of temper-

ature, precipitation and radiation, the 95th percentile of temperature and precipitation, and the number of days on which temperature or precipitation are above or below certain threshold values. The analysis is confined to the five ISIMIP GCMs as this enables a comparison to the same CMIP5 GCM output bias-corrected using the ISIMIP method and the dataset WFD as it was done in ISIMIP. Lastly, a crop model is driven by the different climate input datasets and maize yields are evaluated in terms of both how well they reproduce historical observations and how they are projected to change under one RCP scenario.

The writing style of the manuscript is sloppy and the English is not good. They manuscript contains many figures, which on average have a rather low information content. The discussion of these figures in the text is mostly not very insightful. Comparisons of WFD, WFDEI and EWEMBI data are in many cases unnecessary since it is clear from the definition of these datasets that they are very similar or even identical in many cases. A table that summarizes the differences and commonalities of the datasets for the considered variables should be included in order to make this more transparent to the reader.

The manuscript does not add anything to the existing knowledge on bias correction. New findings are only present in parts of the comparison of the different datasets as well as in the sensitivity analysis of maize yields simulated using these different datasets as input. However, since the latter analysis is done for one crop model in combination with one GCM only, general conclusions cannot be drawn from it. A comparison of its results to corresponding ISIMIP or AgMIP studies might help put the results of this study into context.

In addition to that, the manuscript suffers from a few methodological flaws, see my specific comments below. Overall, I think that a major revision of the manuscript is needed before it can be published in ESD.

**Specific comments**

P2 L6: What are "robust biases"? And what is a "bias" in the context of this paper? I think you should write one or two sentences about that.

P2 L8: "statistical bias-corrections are necessary [...]" – Why? Please explain why you cannot just use the original GCM output as input to your impact models/what would happen if you did so.

P2 L17: "on 4 out of the 5 same CMIP5 GCMs" is not quite correct. Only 3 out of the 4 GCMs chosen in ISIMIP2b were also used in the first phase of ISIMIP.

P2 L22: Which kind of biases have been identified over Central, East and South Africa? Climate models have always been and will always be biased, so merely saying that a model is biased is an empty statement.

P3 L24: Why do you not present results for northern winter and autumn?

P5 L1: How did you interpolate the other variables?

P5 L13: Which WFDEI version do you use? The one with precipitation corrected using CRU (WFDEI-CRU) or GPCC (WFDEI-GPCC) estimates? Please specify.

P5 L22: You should inform the reader here that over land, EWEMBI is identical to WFDEI-GPCC for precipitation, daily mean, minimum and maximum near-surface air temperature and 10 m wind speed. Only for surface downwelling shortwave radiation there is a difference between EWEMBI and WFDEI-GPCC data over land.

P7 L4: Which SRB data exactly do you use for this comparison. Please describe that here or in Sect. 2.2.

Sections 3.1 and 3.2: This is an insufficient description of the CDF-t method. Merely referring the reader to Michelangeli et al. (2009), Déqué (2007) and Vrac et al., (2012, 2016) for all the details is not enough. Please be more specific about how you use $F_{obs,fut}$ to do the quantile mapping in the future period (I assume you map $x$ to $F_{obs,fut}^{-1}(F_{mod,fut}(x))$) and please describe how you estimate $F_{obs,cal}$, $F_{mod,cal}$ and

$F_{mod,fut}$. Are these CDFs estimated parametrically or non-parametrically/empirically and how exactly do you do the estimation? Also I do not understand how you account for seasonality: Do you apply CDF-t month by month or using moving windows? Moreover, you state that "CDF-t preserves any long-term trend in the GCMs data" but do not give any reference that would corroborate that statement. Thinking about Eq. (1), I came to the conclusion that CDF-t does neither preserve trends in moments nor in quantiles. Please discuss this aspect in more detail since the users of your data product will want to know if and how you have modified the trends present in the original CMIP5 GCM data. Lastly, you state that "GCMs data have been interpolated to WFDEI grid before being bias-corrected." Which method do you use for that interpolation?

P8 L3: Not "every GCM has to be calibrated" but the bias-correction method has to be calibrated individually for every GCM.

Sect. 3.3 and Figure S2: I do not understand why you did what you did here. Let's take calibration period 1997–2013 as an example. Did you use 1997–2013 GCM and observational data to calibrate the CDF-t method and then apply the thus calibrated method to the same 1997–2013 GCM data? If that is what you did then I do not understand the purpose of these tests since in that case of course the remaining biases will be small and you cannot draw any conclusion in terms of overfitting. Therefore, also the concluding statement of the section would be nonsense. What you need to do to test for overfitting is a cross-validation.

P9 L2f: What you describe here is not what is shown in Fig. 2.

Figure 3: "Taylor diagrams relative to the mean of surface temperature" does not make sense. Please rephrase. I assume that all panels in the upper row refer to the Sahel box and all panels in the lower row to the Guinean box, correct? Please specify this in the caption. Also, please add a (separate) figure with a map showing the definition of the SAHEL, GUICOAST and any other region used in your study. Lastly, your Taylor diagrams suggest that 1979–2001 JAS mean EWEMBI and WFDEI tas have different

spatial standard deviations despite EWEMBI and WFDEI tas being identical by definition. Can you explain this? Have you maybe used a WFDEI data product version that is different from the one used for the production of EWEMBI? The same question applies to Figures 5, 8, 9, 12, 14, 16.

P18 L1: "daily values" of which variable?

P18 L9f: "CDF-t method is also a bit better than ISIMIP one when one refers to EWEMBI reference data" – of course, because the ISIMIP data were bias-corrected using WFD, your data were bias-corrected using WFDEI data, and EWEMBI is mostly identical to WFDEI (see my other comment above). There is no real point you are making here or elsewhere, where you have made the same statement. Basically, you could leave out the right panels in all Taylor diagram figures related to temperature and precipitation since there is no qualitative difference to the respective middle panel.

P20 L6f: Strange sentence. Please rephrase.

Figure 18: Maps of difference to WFDEI would be good here.

Figure 19: Maps of difference between, say 1970–2000 and 2070–2100 mean values would be good here. For which RCP is this? Do you get qualitatively similar results for the other four ISIMIP GCMs? I mean, why do you do all this work and then just show results for one GCM. . .?

Table 2 and its description in text: Why this selection of sensitivity experiments? Why, for instance, do you do the WFDEIWFDrsds experiment but no WFDEIWFDpr experiment even though the WFDEIpr and WFDEIrsds experiments suggest a higher sensitity of yields to pr than to rsds biases?

P25 L2: "It indicates also that WFD data and related bias-corrected simulations should not be used anymore" – I think your analysis as it currently stands is not sufficient to come to this conclusion because (i) you are comparing potential simulated yields to actual observed yields and (ii) you did not say anything about the quality of the

GDHY data. Nevertheless I think that you are right in terms of rsds as Weedon et al. (2014, doi:10.1002/2014WR015638) report substantial improvements of rsds in WFDEI relative to WFD.

P25 L11ff: "Interannual variability of simulated yields is proportional to the mean with a very weak variability for ISIMIP yield and higher variability for CDF-t and raw simulations. All projections show a clear decrease of maize yields by a factor of 2 over all of West Africa along the 21 st century." – I think you should also show that visually by plotting yields relative to mean 1950–1980 levels, for example.

**Technical corrections**

P2 L10: I think it would be better to write "in combination with" in place of "based on".

P27 L7ff: To be more precise, your list of variables should read "daily accumulated precipitation, daily mean, minimum and maximum near-surface air temperature, daily mean surface downwelling shortwave and daily mean wind speed" here and elsewhere in the manuscript.
* * *

---

## Author Comment (AC1) · 25 Feb 2018

We thank the reviewer for the constructive comments and useful suggestions. Below we answer the different comments of the reviewer. We present all reviewer comments and our answers are given in blue.

General comments:

The presented study developed the bias-corrected CMIP5 GCM daily dataset using a combination of CDF-t method and WFDEI (and EWEMBI) and then compare with the ISI-MIP dataset which based on different bias-correction method and reference data (WFD). Some extreme climate indices (daily-based metrics) as well as maize yield simulated by a crop model were compared across different datasets to characterize the quality of the bias-corrected GCM daily dataset provided in this study. I respect the authors' efforts conducting this comprehensive analysis. Although this study analyzes only West Africa, their findings have implications especially for agricultural impact studies across the world. I only have a few concerns (listed below) and believe that most of them are minor. I recommend the acceptance of this manuscript with minor revision.

Specific comments:

1. Section 3.1. Although the detailed description of CDF-t method may be available in earlier study (Michelangeli et al. 2009, Vrac et al., 2012, 2016), a more completed explanation of key characteristics of the method in this section is unavoidable to make this manuscript stand-alone. Otherwise readers have to scratch around for. Particularly, it would be great if you could add an brief explanation whether the method forces the maximum (or minimum) value of a climatic variable in the future projection to be the same with that in historical period or not.

A more detailed description of the method has been provided with more specific statements and equations. It is indicated that "contrary to the QQ method that projects the GCM CDF of simulated future data onto the CDF of historical data, CDF-t considers the CDF change between GCM historical and future simulations". So the CDF-t method does not force the maximum (or minimum) value of a climatic variable in the future projection to be the same with that in historical period.

2. Section 3.3. I am not convinced whether the current design of the sensitivity analysis is appropriate to evaluate the sensitivity of correction to the length of calibration period. When the data in 1979-1996 are used as the calibration subset, those in

1997-2013 are used as the validation subset; this is fine. But, there is no independent validation subset when the data in 1979-2013 are used as the calibration subset. And it is easily expected that biases in bias-corrected data become the smallest when all available data are used as the calibration data. Therefore, the conclusion that the correction with the longest calibration period leads to the smallest biases is not examined using independent data. However, I think this part is not essential in this study. Removing or reanalysis are possible for this part.

You are right. This conclusion has been removed from the text (and we keep the figures regarding this analysis in Supplementary Information for the readers).

3. Fig. 5. In GUICOAST JAS, ISI-MIP bias-corrected precipitation data distantly distributed from WFD. Why? This is unreasonable because most methods including the ISI-MIP bias-correction method forces GCM data in the historical period similar to the reference data (WFD). A plausible explanation is necessary.

We agree with the reviewer's comment. This figure has been corrected.

4. Fig. 14. Two tendencies are observed in this figure. One is that the 95th percentile precipitation values from ISI-MIP data largely varied by GCM compared to the spread across the GCMs found in CDF-t method. Why?

It is difficult for us to answer this question because we did not applied the ISI-MIP bias-correction method, but only got the simulations from a dedicated website.

Another one is that in SAHEL JAS the IPSL-CM5A-LR data corrected by CDF-t method are relatively far from other GCMs of CDF-t data. A brief explanation is required.

We do not understand this remark as the IPSL-CM5A-LR data corrected by CDF-t is quite near the other bias-corrected GCMs on Fig.14.

5. P25L2. "It indicates also that WFD data and related bias-corrected simulations should not be used anymore." I think this is overstated. Please consider rephrasing or removal. I agree that a use of WFD leads to biased crop yields in crop model simulation mainly due to biases in solar radiation, as demonstrated in your analysis. However, simulated yield variability and/or projected future change in yields would not be affected in relative term when biases in solar radiation are a main issue (for instance, see Iizumi et al., 2010). Reliable projection of yield change in absolute term is challenging, and therefore projected relative change in yield is still only a main source of information for adaptation planning and other application.

You are right. This sentence has been removed.

6. Section 4.3. The presentation of daily-based metrics is relatively not well organized in the current manuscript. The analysis and findings on the daily-based metrics themselves are useful, but not comprehensive compared to earlier study examining daily-based metrics (e.g., Iizumi et al., 2017, JGR, doi:10.1002/2017JD026613). Why did you select a limited number of metrics for this analysis? More importantly, it seems that the importance of the analysis results is not equal to that of the analysis of crop model. The simulated maize yield is used as a metric in the current manuscript, as the daily-based metrics are did so; though the maize yield has much importance in the manuscript compared to the daily-based metrics. A justification to present a limited number of daily-based metrics is necessary.

We chose to select a limited number of daily-based metrics because our main objective in this manuscript is to introduce the new bias-corrected dataset over Africa whose the CDF-t correction method has been applied for the first time. We wanted

also to limit the length of the paper and therefore we focused on priority metrics defined by stakeholders in AMMA-2050 (note that some metrics have been put in Supplementary Information). We have provided now a link to a more complete metrics report produced as a deliverable for AMMA-2050. We chose also to provide a more detailed analysis on the sensitivity of crop simulations to bias-corrected data because this is a quite important issue for stakeholders and farmers, and because we think interesting to evaluate how an impact model can integrate non-linearly the diversity of forcing datasets.

Technical corrections:

7. P1L2-3. Why has CDF-t method never been applied to Africa? Is that due to low availability of daily weather observations in that region?

No, as it is said in the text, CDF-t method has been mainly applied over Europe. It is the first time that Africa has been considered, thanks to AMMA-2050 project. The different and successive reference datasets (WFD, WFDEI, EWEMBI, and others) used in bias-corrections have a global land cover. Of course, daily weather observations are less dense in some areas in Africa, and this introduces a certain level of uncertainty of the reference dataset, that must be kept in mind by users. This is another reason for comparing bias-corrections using various reference datasets.

8. P2L6. "robust" biases. Do you mean "persistent" biases?

Robust biases means biases that have not been reduced up to now, for instance between CMIP3 and CMIP5 GCM simulations.For instance warmer than normal SSTs in the equatorial Atlantic leads to a too southern location of the ITCZ in summer over West Africa. A sentence has been added.

9. P2L19. "high and robust biases". Do you mean "large and persistent biases"?

"Large and robust". We indicate in the text the reference Roehrig et al. (2013).

10. P3L24. "northern summer". Is this "northern hemispheric summer" or "boreal summer"?

We modified for "boreal" summer.

11. P'L7. "anomalies". Probably, "differences" is more appropriate here.

Yes, it is been corrected.

12. Fig. 3. "observations". This should read "references".

Yes, it is been corrected.

13. P12L6. "standardized variance". This should read "standardized standard deviation" by definition of Taylor diagram.

Yes, it is been corrected.

14. P21L6. "GDHY" are a hybrid of FAO country yield data, satellite-derived crop-specific vegetation index and global crop datasets on crop calendar, harvested area and production shares achieved by different growing season. Subnational yield statistics are used to validate the grid-cell yield estimates, but not used as the input to estimate grid-cell yields.

Ok, this has been corrected in the text.

15. P23L9. "Rsds" -> "rsds"?

Yes, this has been corrected.

16. Fig. 19. The label of y-axis should be replaced by "maize yield" instead of "crop yield" to be more precise.

Ok, this has been corrected.

17. Table 2. The caption of the table needs to include sufficient information to interpret the results presented in the table. Saying "see detail in the text" is not acceptable from the viewpoint of readability.

Ok, it has been completed.

All corrections will be applied in the revised manuscript.

---

## Author Comment (AC2) · 25 Feb 2018

We thank the reviewer for the constructive comments and useful suggestions. Below we answer the different comments of the reviewer. We present all reviewer comments and our answers are given in blue.

General comments:

The authors present a dataset of CMIP5 GCM output of daily accumulated precipitation, daily mean, minimum and maximum near-surface air temperature, daily

mean surface downwelling shortwave and daily mean wind speed bias-corrected using the CDFt method and the global gridded observational dataset WFDEI over Western Africa. They compare the raw and the bias-corrected data to WFDEI and 2 other observational datasets, namely WFD and EWEMBI. Unsurprisingly, the bias-corrected data feature much smaller biases relative to WFDEI than the raw data. Specifically, the authors look into biases of seasonal mean values and mean annual cycles of temperature, precipitation and radiation, the 95th percentile of temperature and precipitation, and the number of days on which temperature or precipitation are above or below certain threshold values. The analysis is confined to the five ISIMIP GCMs as this enables a comparison to the same CMIP5 GCM output bias-corrected using the ISIMIP method and the dataset WFD as it was done in ISIMIP. Lastly, a crop model is driven by the different climate input datasets and maize yields are evaluated in terms of both how well they reproduce historical observations and how they are projected to change under one RCP scenario.

The writing style of the manuscript is sloppy and the English is not good. They manuscript contains many figures, which on average have a rather low information content. The discussion of these figures in the text is mostly not very insightful. Comparisons of WFD, WFDEI and EWEMBI data are in many cases unnecessary since it is clear from the definition of these datasets that they are very similar or even identical in many cases. A table that summarizes the differences and commonalities of the datasets for the considered variables should be included in order to make this more transparent to the reader.

Ok. A table summarizing the scores (Correlation, Standard deviation and RMSE) has been added.

The manuscript does not add anything to the existing knowledge on bias correction. New findings are only present in parts of the comparison of the different datasets

as well as in the sensitivity analysis of maize yields simulated using these different datasets as input. However, since the latter analysis is done for one crop model in combination with one GCM only, general conclusions cannot be drawnfrom it. A comparison of its results to corresponding ISIMIP or AgMIP studies might help put the results of this study into context.

Our main objectives of this manuscript are (i) to introduce a new bias-corrected dataset over Africa whose the CDF-t correction method has been applied for the first time, (ii) to quantify the effect of using different reference datasets on the corrected data, (iii) and to illustrate this effect on crop simulations over West Africa. Comparison of these results to corresponding ISPMIP or AgMIP studies is out of the scope of our work, and we do not intend to provide general conclusions regarding crop simulations forced by an ensemble of bias-corrected GCMs.

In addition to that, the manuscript suffers from a few methodological flaws, see my specific comments below. Overall, I think that a major revision of the manuscript is needed before it can be published in ESD.

Specific comments:

P2 L6: What are "robust biases"? And what is a "bias" in the context of this paper? I think you should write one or two sentences about that.

Robust biases means biases that have not been reduced up to now, for instance between CMIP3 and CMIP5 GCM simulations. For instance warmer than normal SSTs in the equatorial Atlantic leads to a too southern location of the ITCZ in boreal summer over West Africa. A sentence has been added.

P2 L8: "statistical bias-corrections are necessary [...]" – Why? Please explain

why you cannot just use the original GCM output as input to your impact models/what would happen if you did so.

We add an explanation: "For instance the too southern ITCZ location in boreal summer over West Africa in most of the GCMs leads to too weak precipitation over the Sahel and to too weak crop yields whose values cannot be used as relevant information for stakeholders and farmers"

P2 L17: "on 4 out of the 5 same CMIP5 GCMs" is not quite correct. Only 3 out of the 4 GCMs chosen in ISIMIP2b were also used in the first phase of ISIMIP.

This has been corrected.

P2 L22: Which kind of biases have been identified over Central, East and South Africa? Climate models have always been and will always be biased, so merely saying that a model is biased is an empty statement.

This has been completed.

P3 L24: Why do you not present results for northern winter and autumn?

This is due to space limit. We have chosen to focus on West Africa as it is the domain concerned in the AMMA-2050 project, and hence to the corresponding boreal summer monsoon season.

P5 L1: How did you interpolate the other variables?

All the variables are indicated here.

[Figure]

P5 L13: Which WFDEI version do you use? The one with precipitation corrected using CRU (WFDEI-CRU) or GPCC (WFDEI-GPCC) estimates? Please specify.

WFDEI-GPCCv5/v6; this has been indicated in the text.

P5 L22: You should inform the reader here that over land, EWEMBI is identical to WFDEI-GPCC for precipitation, daily mean, minimum and maximum near-surface air temperature and 10 m wind speed. Only for surface downwelling shortwave radiation there is a difference between EWEMBI and WFDEI-GPCC data over land.

Ok, this has been added.

P7 L4: Which SRB data exactly do you use for this comparison. Please describe that here or in Sect. 2.2.

SRB release 3.0; this has been added.

Sections 3.1 and 3.2: This is an insufficient description of the CDF-t method. Merely referring the reader to Michelangeli et al. (2009), Déqué (2007) and Vrac et al., (2012, 2016) for all the details is not enough. Please be more specific about how you use Fobs,fut to do the quantile mapping in the future period (I assume you map x to $F^{-1}$obs,fut(Fmod,fut(x))) and please describe how you estimate Fobs,cal, Fmod,cal and Fmod,fut. Are these CDFs estimated parametrically or non-parametrically/empirically and how exactly do you do the estimation? Also I do not understand how you account for seasonality: Do you apply CDF-t month by month or using moving windows? Moreover, you state that "CDF-t preserves any long-term trend in the GCMs data" but do not give any reference that would corroborate that statement. Thinking about Eq. (1), I came to the conclusion thatCDF-t does neither preserve trends in moments nor in quantiles. Please discuss this aspect in moredetail

since the users of your data product will want to know if and how you have modified the trends present in the original CMIP5 GCM data. Lastly, you state that "GCMs data have been interpolated to WFDEI grid before being bias-corrected." Which method do you use for that interpolation?

A more detailed description of the method has been provided with more specific statements and equations. These CDFs are estimated non parametrically. Seasonality has been taken into account by performing month-by-month corrections. Regarding CDF-t preservation of the long-term trend in GCM data, a reference has been added, as well as a comment stating that CDF-t does neither preserve trends in moments nor in quantiles. The interpolation methods have been indicated too.

P8 L3: Not "every GCM has to be calibrated" but the bias-correction method has to be calibrated individually for every GCM.

Ok, this has been corrected.

Sect. 3.3 and Figure S2: I do not understand why you did what you did here. Let's take calibration period 1997–2013 as an example. Did you use 1997–2013 GCM and observational data to calibrate the CDF-t method and then apply the thus calibrated method to the same 1997–2013 GCM data? If that is what you did then I do not understand the purpose of these tests since in that case of course the remaining biases will be small and you cannot draw any conclusion in terms of overfitting. Therefore, also the concluding statement of the section would be nonsense. What you need to do to test for overfitting is a cross-validation.

You are right. This conclusion about over-fitting has been removed from the text. We choose to keep the longest period to perform the calibration period. As Referee 1 says, it is not essential to go further regarding this part.
P9 L2f: What you describe here is not what is shown in Fig. 2.

This has been corrected.

Figure 3: "Taylor diagrams relative to the mean of surface temperature" does not make sense. Please rephrase. I assume that all panels in the upper row refer to the Sahel box and all panels in the lower row to the Guinean box, correct? Please specify this in the caption. Also, please add a (separate) figure with a map showing the definition of the SAHEL, GUICOAST and any other region used in your study. Lastly, your Taylor diagrams suggest that 1979–2001 JAS mean EWEMBI and WFDEI tas have different spatial standard deviations despite EWEMBI and WFDEI tas being identical by defi- nition. Can you explain this? Have you maybe used a WFDEI data product version that is different from the one used for the production of EWEMBI? The same question applies to Figures 5, 8, 9, 12, 14, 16.

The caption has been corrected and SAHEL-GUICOAST boxes have been superimposed on Figures 1, 18 and 19.
We used the same version of dataset for WFDEI and EWEMBI. All relevant figures have been modified.

P18 L1: "daily values" of which variable?

This has been corrected.

P18 L9f: "CDF-t method is also a bit better than ISIMIP one when one refers to EWEMBI reference data" – of course, because the ISIMIP data were bias-corrected using WFD, your data were bias-corrected using WFDEI data, and EWEMBI is mostly identical to WFDEI (see my other comment above). There is no real point you are

making here or elsewhere, where you have made the same statement. Basically, you could leave out the right panels in all Taylor diagram figures related to temperature and precipitation since there is no qualitative difference to the respective middle panel.

We prefer to keep these diagrams because it enables to show more directly the proximity of the WFD and WFDEI bias-corrected data to EWEMBI than when WFDEI is used as the reference in the Taylor diagrams.

P20 L6f: Strange sentence. Please rephrase.

It has been corrected.

Figure 18: Maps of difference to WFDEI would be good here.

We added the map of the difference between EWEMBI and WFDEI associated crop yields.

Figure 19: Maps of difference between, say 1970–2000 and 2070–2100 mean values would be good here. For which RCP is this? Do you get qualitatively similar results for the other four ISIMIP GCMs? I mean, why do you do all this work and then just show results for one GCM. . .?

The map of difference has been added. This is RCP8.5 scenario (it has been indicated in the text and the caption). We have not run the simulations with the other ISMIP GCMs because it is out of the scope of our work.We chose to focus on sensitivity experiments carried out on one GCM. We do not intend to provide general conclusions regarding crop simulations forced by an ensemble of bias-corrected GCMs.

Table 2 and its description in text: Why this selection of sensitivity experiments? Why, for instance, do you do the WFDEIWFDrsds experiment but no WFDEIWFDpr experiment even though the WFDEIpr and WFDEIrsds experiments suggest a higher sensitivity of yields to pr than to rsds biases?

We have chosen to show the sensitivity to (i) first the reference data sets (WDF, WFDEI, EWEMBI), (ii) then to the IPSL GCM non-corrected data and data corrected with WFD and WFDEI reference dataset, (iii) to one out of the WFDEI variables by replacing it by the corresponding raw IPSL-CM5A-LR data, (iv) at last to rsds from WFD instead of WFDEI. It is a rather extensive set of simulations. We performed the last sensitivity experiment because the difference in rsds between WFD and WFDEI is quite high (see Fig.1g-h-j-k) while the difference in pr is quite weak (see Fig.1d-e).

P25 L2: "It indicates also that WFD data and related bias-corrected simulations should not be used anymore" – I think your analysis as it currently stands is not sufficient to come to this conclusion because (i) you are comparing potential simulated yields to actual observed yields and (ii) you did not say anything about the quality of the GDHY data. Nevertheless I think that you are right in terms of rsds as Weedon et al. (2014, doi:10.1002/2014WR015638) report substantial improvements of rsds in WFDEI relative to WFD.

You are right. This sentence has been removed.

P25 L11ff: "Interannual variability of simulated yields is proportional to the mean with a very weak variability for ISIMIP yield and higher variability for CDF-t and raw simulations. All projections show a clear decrease of maize yields by a factor of 2 over all of West Africa along the 21 st century." – I think you should also show that visually by plotting yields relative to mean 1950–1980 levels, for example.

These plots have been added.

Technical corrections:

P2 L10: I think it would be better to write "in combination with" in place of "based on".

This has been corrected.

P27 L7ff: To be more precise, your list of variables should read "daily accumulated precipitation, daily mean, minimum and maximum near-surface air temperature, daily mean surface downwelling shortwave and daily mean wind speed" here and elsewhere in the manuscript.

This has been corrected.

All corrections will be applied in the revised manuscript.